# Predicting the viability of beta-lactamase: How folding and binding free energies correlate with beta-lactamase fitness

Jordan Yang[1], Nandita Naik[2], Jagdish Suresh Patel[3,4], Christopher S. Wylie[2], Wenze Gu[1], Jessie Huang[5], F. Marty Ytreberg[3,6], Mandar T. Naik[7], Daniel M. Weinreich[2], Brenda M. Rubenstein[1] *

1 Department of Chemistry, Brown University, Providence, Rhode Island, United States of America, 2 Department of Ecology and Evolutionary Biology, Brown University, Providence, Rhode Island, United States of America, 3 Institute for Modeling Collaboration and Innovation, University of Idaho, Moscow, Idaho, United States of America, 4 Department of Biological Sciences, University of Idaho, Moscow, Idaho, United States of America, 5 Department of Chemistry, Wellesley College, Wellesley, Massachusetts, United States of America, 6 Department of Physics, University of Idaho, Moscow, Idaho, United States of America, 7 Department of Molecular Pharmacology, Physiology, and Biotechnology, Brown University, Providence, Rhode Island, United States of America

* brenda_rubenstein@brown.edu

**Data Availability Statement:** All input/output, script, and data files are available from this manuscript's GitHub Repository,

## Abstract

One of the long-standing holy grails of molecular evolution has been the ability to predict an organism's fitness directly from its genotype. With such predictive abilities in hand, researchers would be able to more accurately forecast how organisms will evolve and how proteins with novel functions could be engineered, leading to revolutionary advances in medicine and biotechnology. In this work, we assemble the largest reported set of experimental TEM-1 $\beta$-lactamase folding free energies and use this data in conjunction with previously acquired fitness data and computational free energy predictions to determine how much of the fitness of $\beta$-lactamase can be directly predicted by thermodynamic folding and binding free energies. We focus upon $\beta$-lactamase because of its long history as a model enzyme and its central role in antibiotic resistance. Based upon a set of 21 $\beta$-lactamase single and double mutants expressly designed to influence protein folding, we first demonstrate that modeling software designed to compute folding free energies such as FoldX and PyRosetta can meaningfully, although not perfectly, predict the experimental folding free energies of single mutants. Interestingly, while these techniques also yield sensible double mutant free energies, we show that they do so for the wrong physical reasons. We then go on to assess how well both experimental and computational folding free energies explain single mutant fitness. We find that folding free energies account for, at most, 24% of the variance in $\beta$-lactamase fitness values according to linear models and, somewhat surprisingly, complementing folding free energies with computationally-predicted binding free energies of residues near the active site only increases the folding-only figure by a few percent. This strongly suggests that the majority of $\beta$-lactamase's fitness is controlled by factors other than free energies. Overall, our results shed a bright light on to what extent the community is justified in using thermodynamic measures to infer protein fitness as well as how applicable

yanghaobojordan/PLOS_BetaLactamase: https://github.com/yanghaobojordan/plos_betalactamase.

**Funding:** JY, NN, JSP, WG, JH, MY, MTN, DMW, and B MR were funded by National Science Foundation EPSCoR Track-II award number OIA1736253. JSP was additionally supported by the Center for Modeling Complex Interactions sponsored by the NIGMS under award number P20 GM104420. Computer resources employed by JY, JH, WG, and BMR were provided by the Brown Center for Computation and Visualization (CCV) and JSP and MY employed resources provided by the high-performance computing center at Idaho National Laboratory, which is supported by the Office of Nuclear Energy of the U.S. DOE and the Nuclear Science User Facilities under Contract No. DE-AC07-05ID14517. EPSCOR Grant Information: https://www.nsf.gov/awardsearch/showAward?AWD_ID=1736253 Center for Modeling Complex Interactions Information: http://grantome.com/grant/NIH/P20-GM104420-04 No, the funders had no role in study design, data collection and analysis, decision to publish, or preparation of the manuscript.

**Competing interests:** No authors have competing interests.

modern computational techniques for predicting free energies will be to the large data sets of multiply-mutated proteins forthcoming.

## Introduction

The ability to predict how an organism's fitness is influenced by mutations is central to being able to project and, in some cases, steer the course of natural evolution [1–3], engineer protein sequences with novel biological functions [4, 5], and treat genetic disorders [6]. Nevertheless, to this day, such predictions remain far from routine. Only in rare instances can a mutation's effect on an organism's fitness be directly tied to a single phenotypic consequence, such as a protein's fitness for performing a specific function. Yet, even in those rare instances, even the simplest protein's fitness is influenced by a wide variety of factors [7] including protein and gene expression levels [8], interactions with chaperones [9–11], protein folding stability [12–15], protein folding dynamics [16, 17], and proteolytic susceptibility [18]—as well as many complex factors yet to be uncovered or understood. Unfortunately, many of these even well-understood factors are often difficult, if not impossible, to model *in vitro* or *in silico* [19], limiting their overall utility. Given this backdrop, simple, calculable indicators that can predict phenotypes, and ultimately, organismal fitness, are of high value and in high demand.

One experimentally accessible set of phenotypic predictors for the effects of nonsynonymous mutations are protein biophysical measures, such as proteins' thermodynamic stabilities [6, 14, 20, 21]. As nearly all biological processes and structures involve proteins, one would expect that proteins' abilities to properly fold, catalyze small-molecule substrates, or bind to partners are highly correlated with their proper function, and by extension, organisms' abilities to survive and reproduce. How these abilities are influenced by mutations may be quantified by the thermodynamic predictors $\Delta\Delta G_{fold}$, the change in a protein's folding free energy upon mutation relative to the wild type, and $\Delta\Delta G_{bind}$, the change in a protein's binding free energy for a given substrate upon mutation relative to the wild type. Negative $\Delta\Delta G$ values indicate that proteins are stabilized by a mutation, while positive values indicate that they are destabilized (see the Supporting Information for further details). Most proteins have an optimal thermodynamic regime in which they function, as being too stable can also compete with their ability to function [22]. Indeed, past research has shown that most globular proteins have $\Delta G_{fold}$ values in the range of -5 to -15 kcal/mol and that most mutations are accompanied by $\Delta\Delta G_{fold}$ values of -4 to 10 kcal/mol, meaning that many mutations possess $\Delta G_{fold}$ values roughly equal to zero and therefore exist at the edge of stability [23]. $\Delta\Delta G_{fold}$ values may be experimentally determined via circular dichroism [24], differential scanning calorimetry [25, 26], or single-molecule fluorescence techniques [27], while $\Delta\Delta G_{bind}$ values may be determined via isothermal titration calorimetry [28] or surface plasmon resonance [29]. Although advances in saturation mutagenesis for producing a plurality of mutations [30] and deep sequencing for rapid sequencing large numbers of mutants [31] have accelerated aspects of these techniques, measuring protein free energy changes remains a comparatively low-throughput and time-intensive process, largely owing to the time it takes to express and purify hundreds to thousands of proteins. Thus, while past Herculean experiments on single mutants have produced a smattering of free energy data [32–37] and more recent quasi-exhaustive approaches have shed light on distributions of fitness effects by directly measuring fitness [38–41], even higher throughput means of estimating the free energy changes of proteins' full

complement of mutations are needed to forge a more complete picture of the correlation between biophysical predictors and organismal phenotypes.

A key tool that has emerged for accelerating the estimation of these predictors is computation. Thermodynamic quantities such as the free energies discussed above may be calculated via equilibrium statistical mechanical simulations of the underlying proteins. While molecular dynamics [42, 43] and Monte Carlo [44] simulations that attempt to fully sample proteins' degrees of freedom based upon judiciously parameterized force fields are most accurate for estimating these quantities, these simulations are often orders of magnitude too slow to separately model each of a protein's thousands of distinct single, nevermind multiple, mutants. Indeed, conventional molecular dynamics simulations of just a handful of mutants remains state-of-the-art [45]. What has therefore transformed the field by making the prediction of free energies of large numbers of mutations not only viable, but routine, is the development of empirical effective free energy function techniques [46, 47], which take in the conformations of proteins and ligands, and directly estimate their $\Delta\Delta G_{fold}$ and $\Delta\Delta G_{bind}$ values using functions parameterized on large databases of protein free energies. Such simulations have enabled a number of previously inconceivable comparisons between mutant free energy changes and organismal measures of fitness, such as minimum inhibitory concentrations (MIC) in bacteria [39] or the viability of viral plaques [15]. One of the primary messages to arise from these studies has been that fitness often falls off precipitously as a proteins' $\Delta G$ value surpasses 0 and therefore that large, positive $\Delta\Delta G$ values correlate with low fitness, but not necessarily vice-versa [6, 48]. Despite these seminal findings, much remains to be understood not only about the accuracy with which empirical free energy functions predict individual proteins' free energy changes upon mutation, but the finer relationships between free energies and fitness.

In this work, we experimentally determine the folding free energies of 21 TEM-1 $\beta$-lactamase single and double mutants and compare them with computational results for the $\Delta\Delta G$ values of folding from a variety of empirical free energy function techniques. As our data set of experimental $\beta$-lactamase folding free energies is the largest currently available, it has granted us the unique opportunity to make apples-to-apples comparisons between computational and experimental folding free energies, unlike previous works which have been constrained to apples-to-oranges comparisons of folding free energies to fitness [38, 49]. We then analyze how predictive these experimental and computational free energies are of TEM-1 $\beta$-lactamase fitness. The TEM-1 $\beta$-lactamase protein is a model enzyme [50] that hydrolyzes such essential $\beta$-lactam drugs as penicillins (including the ampicillin modeled here) and cephalosporins, and is therefore directly responsible for the evolution of many common forms of bacterial drug resistance (see the Supporting Information for further information about $\beta$-lactamases, including TEM-1) [51, 52]. Beyond basic research interest, being able to predict the fitness of TEM-1's many possible mutants is therefore also helpful for predicting and ultimately combating the mutants that will lead to the next-generation of drug-resistant, "superbug" bacteria. To compute $\Delta\Delta G$ values of folding and binding, we employ FoldX (and MD+FoldX) [53], PyRosetta [54], PoPMuSiC [55], and AutoDock Vina[1]Please note that PyRosetta and AutoDock Vina utilize a combination of empirical and physical free energy contributions by weighting physically-inspired terms based upon fits to larger data sets. They are therefore not strictly empirical free energy function techniques. [56]. These programs were selected from numerous possible packages [57] because of the balance of computational expediency and accuracy they bring to the problem of predicting single mutant free energies. We find that PyRosetta, in particular, accurately reproduces the experimental folding free energies of the single, and less methodically, double mutants studied. Using these reasonably accurate single-mutant free energies of folding, we then studied how correlated folding free energies are with $\beta$-lactamase fitness. We demonstrate that the overall low predictive capacity of folding free energies alone can be

boosted by supplementing them with information about binding free energies. Nevertheless, as may be expected given the complexity of the overall transcription, translation, and post-translation processes, we demonstrate that thermodynamic descriptors only explain a small fraction of *β*-lactamase fitness results. Overall, our findings shed light not only on the accuracy of high-throughput approaches for estimating protein thermodynamic predictors, but also on just how predictive of organismal fitness these measures can be anticipated to be for an important model protein.

## Materials and methods

### Experimental determination of folding free energies

We began by experimentally determining the folding free energies of 21 TEM-1 *β*-lactamase mutants, informally known as the 'Wylie' mutants (see Fig 1 for a listing and Fig 2 for an illustration of the relative locations of the Wylie mutants), and wild type TEM-1 using circular dichroism.

In order to do so, the Wylie mutants were first sub-cloned into a pBAD202 expression vector available in the pBAD directional TOPO expression kit (Invitrogen). The native leader peptide sequence of the *β*-lactamase gene was retained to achieve periplasmic transport for proper folding of the enzyme. Since TEM-1 has a weak intrinsic affinity for metal ions, no purification tag was added to the protein. The plasmids carrying the gene of each mutant were then transformed into TOP10 (Invitrogen) *E. coli* strains and 25 ml LB media starter cultures were

| Wylie Mutant Data Set | | |
|---|---|---|
| **Single Mutants** | | |
| A172P | G218V | L199F |
| A213G | G283C | R93S |
| D163Y | I142F | R241H |
| E212K | K234Q | R275G |
| G144E | L57H | S70G |
| **Double Mutants** | | |
| A172P/G283C | D163Y/R93S | G144E/L199F |
| A213G/L57H | E212K/G218V | K234Q/R241H |

| Method | Data Set |
|---|---|
| Circular Dichroism for $\Delta\Delta G_{fold}$ | Wylie Mutants |
| FoldX for $\Delta\Delta G_{fold}$ | All Single, Select Higher–Order Mutants |
| PyRosetta for $\Delta\Delta G_{fold}$ | Wylie Mutants |
| PopMusic for $\Delta\Delta G_{fold}$ | All Single Mutants |
| Autodock Vina for $\Delta\Delta G_{bind}$ | All Mutants < 8 Å from the Active Site |

**Fig 1.** **(Top)** Table of the 15 single and 6 double TEM-1 mutants that constitute the Wylie mutant data set. Note that in naming the mutants, we first specify the wild type residue abbreviation, followed by the Ambler residue number, and end with the mutant residue abbreviation. **(Bottom)** The mutant data sets produced or analyzed using the different experimental and computational methods described in this manuscript.

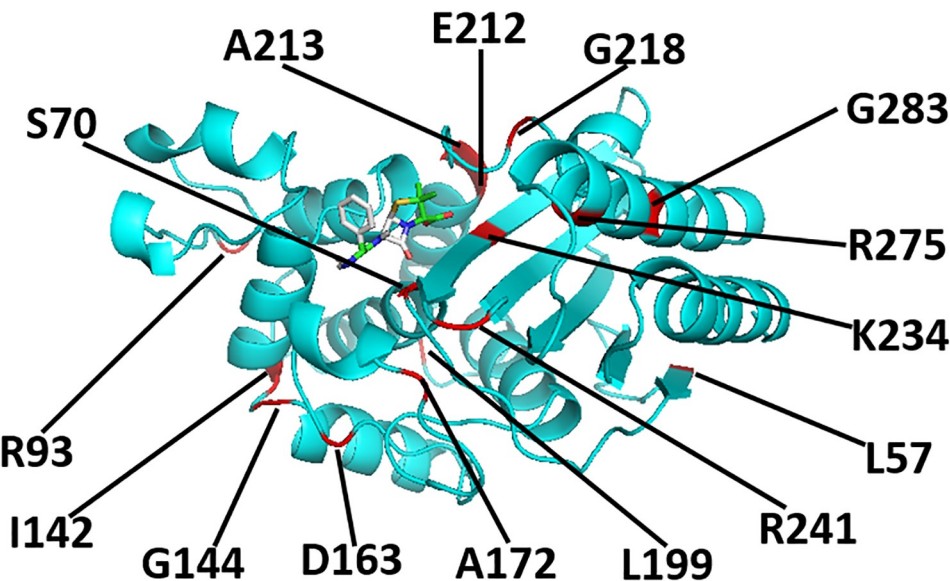

**Fig 2. A cartoon representation of the TEM-1 $\beta$-lactamase protein (PDB ID: 1xpb) and its ampicillin ligand after the wild type protein has been relaxed and ampicillin docked to it.** The positions of many of the residues mutated in this work, including S70 and G144, are indicated in red.

grown overnight at 37˚C using kanamycin as a selection marker. The next morning, the cells were transferred to 800 ml of fresh LB culture supplemented with kanamycin and protein induction was achieved by the addition of 0.1% arabinose at culture $OD_{600 \sim}$ 0.6. The culture was then grown overnight at 18˚C and spun at 4000 rpm in a Sorvell RC5 centrifuge. The supernatant medium was next discarded and the cell pellet was suspended in sucrose buffer (30 mM TRIS, 20% Sucrose; pH 8.0). The suspension was spun at 6000 rpm and the supernatant was again discarded. The resultant cell pellet was subsequently gently resuspended in $MgSO_4$ buffer (5 mM $MgSO_4$; pH 7.0) to induce osmotic shock and incubated at 4˚C for 30 minutes for maximum release of the enzyme from the periplasm. The protein was separated from the cells by spinning at 14000 rpm. This periplasmic extract was then incubated with 5 ml Ni-NTA beads (Qiagen) at 4˚C for 15 minutes and the resin slurry was packed on an open chromatography column. The flow-through from the column was discarded and the column was generously washed with binding buffer (50 mM potassium phosphate, 100 mM NaCl; pH 7.5). The protein was eluted using 10 column volumes of elution buffer (50 mM potassium phosphate, 100 mM NaCl, 15 mM imidazole; pH 7.5). The sample was afterwards concentrated to < 5 ml using an Amicon Ultra-15 centrifugation device with a membrane with a 10 KDa molecular weight cutoff. The sample was then passed through a Superdex-75 16/600 size exclusion column connected to a GE Akta FPLC using the storage buffer (200 mM potassium phosphate, 4% glycerol, pH 7.0). The purity of these samples was ascertained using SDS-PAGE and the protein concentration was determined using 280 nM absorbance with an UV spectrophotometer. The samples were flash frozen in liquid nitrogen and stored at -80˚C. Our typical yields were 2–20 mg purified $\beta$-lactamase per liter of LB media.

The thermodynamic stability of each allele was determined by circular dichroism (CD) on a Jasco J-815 spectrometer in a 200 mM potassium phosphate pH 7.0 with 4% glycerol buffer. Briefly, 15 $\mu$M of each enzyme was subjected to increasing temperature at the rate of 2˚C/min in 2 mm cuvettes. The relatively slow temperature ramp and small cuvettes helped to ensure that the samples attained equilibrium at each temperature. Changes in the ellipticity at 223 nM

were recorded from 20 to 90˚C. The experiment was performed in triplicate and the melting temperature ($T_m$) and van't Hoff enthalpy ($\Delta H$) were calculated by fitting resultant data to a two-state transition as described in previous work (see S1 Table in S1 File for a summary of the circular dichroism data obtained) [24, 50].

## Computing free energies of folding with FoldX

As a computational starting point, the folding free energies ($\Delta\Delta G_{fold}$) of *all* 4978 (= 262 residues × 19 possible mutations per residue) TEM-1 $\beta$-lactamase single missense mutants and specific double mutants were calculated using the FoldX 4.0 algorithm [47, 53]. FoldX was selected to accomplish this necessarily high throughput task due to its relatively high accuracy among fast algorithms—the algorithm has been demonstrated to achieve correlation coefficients as large as 0.7 on a mix of ProTherm [58] and the 1088 Guerois mutants [47], outperforming several alternative algorithms based on both empirical and physical force fields [57]—at minimal computational expense. Of relevance to this work, FoldX is especially designed to model *single* mutants and has been trained on the select set of $\beta$-lactamase mutants found in the ProTherm database, but has been infrequently applied to multi-point mutants [59].

In order to obtain folding free energy differences for $\beta$-lactamase, we initialized our FoldX calculations with the 1xpb Protein Data Bank $\beta$-lactamase structure [60]. The structure file was first modified to remove everything but TEM-1 $\beta$-lactamase and crystal structure water molecules. To match the experimental residue numbering, the residues between 51 and 58 were renumbered sequentially.

FoldX simulations were performed on structures with and without prior molecular dynamics relaxation of the initial wild type conformation. In the following, we term those simulations in which molecular dynamics relaxation was performed before FoldX free energies were computed MD+FoldX simulations. Past studies performed by our team have shown that relaxing the wild type structure before introducing mutations can significantly improve the predictive capacity of FoldX on proteins, such as TEM-1, on which FoldX was not explicitly trained [61, 62]. In our MD+FoldX simulations, the final clean structure file was used to carry out atomistic molecular dynamics simulations using the protocol reported in our previous studies [61, 62]. Briefly, the GROMACS 2018.4 software package was used to perform the MD simulations with the AMBER99SB*-ILDN force field [63]. Final production simulations were carried out for 100 ns and snapshots were preserved every 1 ns resulting in 100 snapshots of each TEM-1 $\beta$-lactamase structure. Either the MD snapshots or cleaned PDB structures were then repaired using the RepairPDB function six times in succession to minimize and converge the potential energy [64]. During this procedure, FoldX searches for residues with poor torsion angles due to incorrect rotamer assignment, and after calculating interactions with neighboring atoms, replaces them with the correct rotamer assignment. FoldX subsequently performs a local optimization of the side chains to mitigate van der Waals interactions. Lastly, FoldX identifies residues with high free energies and samples new rotamer combinations composed of these residues and their neighbors to pinpoint new free energy minima.

After optimizing the original 3D structures, all mutant structures were generated using the FoldX BuildModel command [64]. Subsequently, FoldX selects the rotamer with the optimal placement. Mutant free energy changes are lastly calculated based upon these final structures using the FoldX free energy function [53]. Given their previous success reproducing experimental free energy changes [57], in this work, the FoldX weights in its free energy function were set to their default values. In order to compute $\Delta\Delta G$ values, the difference between the $\Delta G$ of each mutant and the wild type was taken. In many cases, this leads to fortuitous error cancellations, particularly involving difficult to evaluate free energies of the unfolded proteins,

that improve the overall accuracy of the predictions. For the MD+FoldX calculations, the final $\Delta\Delta G_{fold}$ values for each mutant were obtained by averaging the FoldX results across all individual snapshot estimates.

## Computing free energies of folding with PyRosetta

In order to improve upon the accuracy of our FoldX calculations, we also employed PyRosetta to compute free energies of folding. PyRosetta is an independent Python-based implementation of the Rosetta molecular modeling package that enables users to design and implement structure prediction and design algorithms using its underlying Rosetta sampling and scoring functions [54]. Because PyRosetta possesses more robust ways of relaxing mutant structure side chains than FoldX, it is expected to yield more accurate predictions than FoldX, particularly for mutants in which more compact or weakly charged wild type residues are substituted with more voluminous or highly charged mutant residues. Recent work by Kellogg *et al.* has demonstrated that PyRosetta can achieve correlation coefficients in excess of 0.5 against databases of experimental folding free energies [65].

In this work, we used PyRosetta-4 [54] to capture the difference in Rosetta score directly in experimentally comparable units of kcal/mol [66] between each mutant structure and wild type TEM-1 represented by the PDB 1xpb structure [60]. During our PyRosetta simulations, we first repacked all 1xpb side chains by sampling from the 2010 Dunbrack rotamer library [67], and then used Monte Carlo (MC) sampling coupled with energy minimization to optimize the wild type structure based upon the Rosetta REF2015 scoring function [66]. Next, we introduced each missense mutation and repacked all residues within a 10 Å distance of the mutated residue's center, followed by a linear minimization of the backbone and all side chains. As part of our protocol, we performed five independent simulations of 300,000 Monte Carlo cycles each, and the predicted $\Delta\Delta G_{fold}$ value was taken to be the average of the two lowest scoring structures of the five. During each simulation, each mutant structure was perturbed, and accepted or rejected based upon the Metropolis criterion. We selected a 10 Å repacking radius as it served as a reasonable compromise between achieving accurate relaxation without getting caught in metastable minima and computational expediency [38], although other radii could in principle be selected. Because PyRosetta performs Monte Carlo sampling and minimization, it is significantly more computationally expensive than FoldX. We therefore primarily applied it to the Wylie mutant data set and mutants for which FoldX predictions seemed questionable relative to other experimental and simulation predictions.

## Computing ampicillin binding free energies with AutoDock Vina

As residues nearest the active site are expected to most influence binding free energies (for exceptions, see Stiffler *et al.* [41]), ampicillin docking simulations were performed on residues within an 8 Å distance of the active site. Ampicillin was chosen as a substrate so as to be consistent with the ampicillin-based minimum inhibitory concentration (MIC) and fitness experiments described below. Docking was performed using AutoDock Vina (ADV) 1.1.2 [56]. As with PyRosetta, AutoDock uses physically-inspired scoring functions that intake protein and ligand conformations weighted to reproduce experimental binding free energies [56]. With this scoring function, ADV determines the lowest free energy protein-ligand conformations by taking a sequence of steps consisting of a proposed change in conformation followed by an optimization performed by the Iterated Local Search global optimizer algorithm [68, 69]. The performance of the AutoDock and ADV scoring functions were compared with that of 29 other functions during the CASF-2013 benchmark, which assessed the performance of scoring functions [70]. These benchmarks illustrated that ADV outperforms AutoDock and 75% of

other tested methods. ADV therefore represents a useful compromise between speed and accuracy for the purposes of high-throughput calculations.

For our ADV simulations, the ligand PDBQT input file was prepared by first downloading the ampicillin pdb file from PubChem (https://pubchem.ncbi.nlm.nih.gov/compound/6249). Because of ampicillin's low pKa of 2.5, all ampicillin carboxylic acid groups were modeled as carboxylates in our ADV simulations. 6 out of 32 bonds in the ligand were made rotatable. Then, all hydrogens were added to the ligand using AutoDockTools, Gasteiger charges were computed, and the non-polar hydrogens were merged. As there is no co-crystal of ampicillin with TEM-1 publicly available, before performing docking calculations on mutants, preliminary docking calculations had to first be performed on wild type TEM-1 to identify a reliable ampicillin docked pose (see S9 Fig in S1 File). After finding that pose, all 12 residues that were within an 8 Å distance of the alpha carbon of residue 70 (representative of the active site) were selected for binding affinity calculations. First, PyRosetta was used to introduce each missense mutation and relax each mutant structure. Then, to prepare each receptor PDBQT file, Auto-DockTools was used to first add polar hydrogens to the macromolecule and then to assign Kollman United Atom charges. To create a configuration file, a grid box with a size of 24 Å × 26 Å × 24 Å was generated and centered on the $\alpha$-carbon of residue 70. We chose to employ a relatively small grid box so as to reduce the chance that ampicillin binds to a non-active site region of the enzyme, which is undesirable. We set *num_modes* to 5 and *exhaustiveness* to 8. The remaining docking parameters were kept at their default values. At the conclusion of each docking calculation, the predicted binding affinity in kcal/mol of the best mode was selected among the 5 generated binding modes and reported in the figures below.

## Experimental measures of fitness

There are numerous ways of characterizing organismal fitness, even within the same organism. As this study focuses on mutations to the TEM-1 enzyme that confer antibiotic resistance, we have chosen to gauge the fitness of bacteria containing TEM-1 mutants based upon how resistant they remain to one of the primary $\beta$-lactam antibiotics, ampicillin. This resistance may be quantified by minimum inhibitory concentrations (MIC), which are the lowest concentrations of, in this case, ampicillin, that prevent all detectable bacterial growth. While we have determined the MIC values of the Wylie mutant data set (see the Supporting Information, including S6 and S7 Figs in S1 File, for further information), here, we overwhelmingly employ the more comprehensive set of fitness values acquired by Firnberg *et al.* in our analyses (see Fig 1) [38]. Firnberg *et al.* quantified each mutant's fitness by taking an average of the number of copies of the mutant alleles weighted by the range of ampicillin concentrations at which they were grown and normalizing this by the wild type average [38]. We have verified that the Firnberg fitness values correlate well ($r^2 > 0.7$) with our previously determined MIC values as well as other data sets we have acquired, thus validating their use here (see S6 Fig in S1 File).

## Results

### The Wylie mutant data set: Direct comparisons between computational and experimental free energies of folding

**Accuracy of single mutant predictions.**   In order to first assess how accurately computational techniques predict the $\Delta\Delta G_{fold}$ values of $\beta$-lactamase, we begin by comparing our experimentally-determined folding free energies to our computational predictions from both MD +FoldX and PyRosetta. Note that throughout the remainder of this paper, 'free energy changes' will refer to $\Delta\Delta G$ values for brevity. Because experimentally determining folding free energies

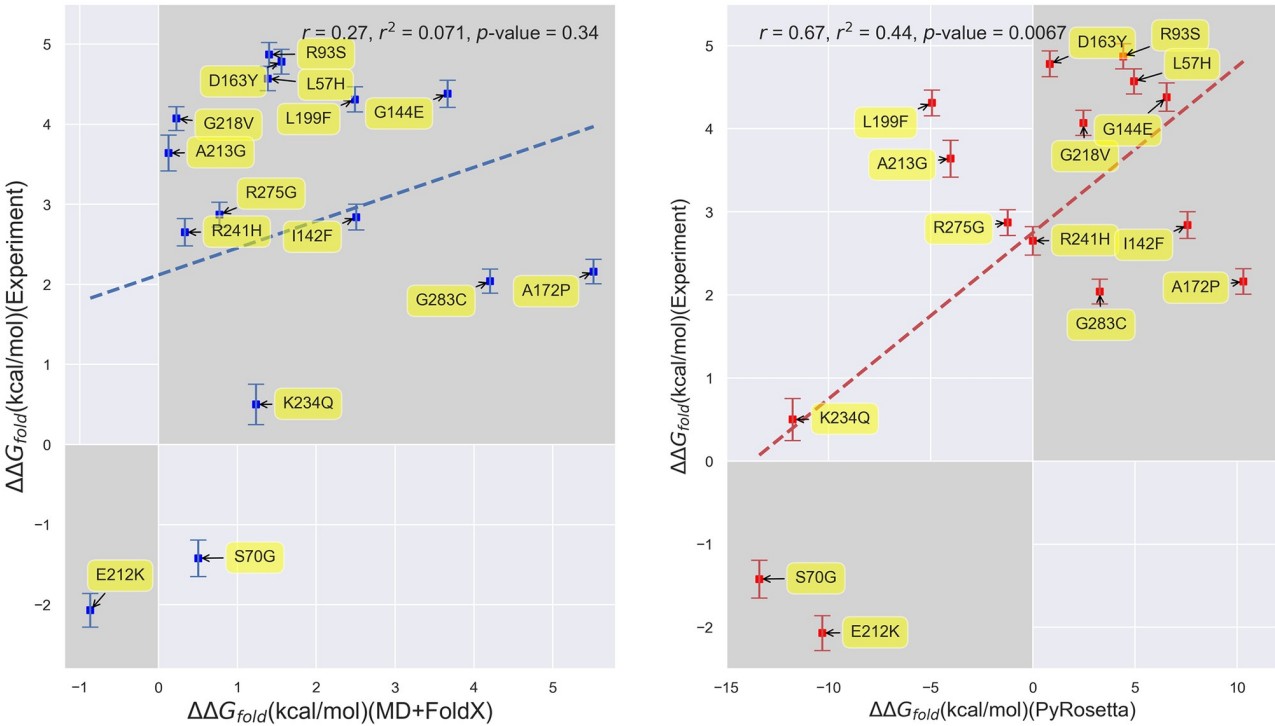

**Fig 3. Scatterplots depicting the correlation between experimentally- and computationally-determined β-lactamase TEM-1 free energies of folding for single mutants. (Left)** Correlation based upon MD+FoldX free energy predictions. **(Right)** Correlation based upon PyRosetta free energy predictions. The labels on the points indicate the mutant name based upon residue changes to the wild type structure as described above. The MD +FoldX coefficient of determination is $r^2 = 0.071$, which is significantly lower than the $r^2 = 0.44$ value obtained using PyRosetta. The shaded regions delineate the first and third quadrants of the plot, which contain mutants whose free energies are of the same sign according to both experiment and computation.

is inherently low throughput, we focused our experimental efforts on the Wylie mutant data set (see Fig 1 for a detailed list of these mutants). The residues within this set are all greater than 6 Å from the active site and were purposefully selected because, at these distances, they were expected to have potentially significant effects on folding, but limited effects on binding and kinetics, allowing us to mostly attribute their influence on fitness to folding changes. The one exception is the S70G mutant, as residue 70 resides in the heart of the binding site and transforming β-lactamase's primary catalytic serine into an inert glycine is known to markedly decrease the enzyme's catalytic efficiency, yet markedly increase its folding stability [71, 72]. In Fig 3, we plot experimentally-determined folding free energies against computationally-predicted free energies for the *single* mutants. We have shaded the first and third quadrants in this figure to ease identification of the mutants whose experimental and computational free energies are of the same sign. It is thus gratifying to see that both MD+FoldX and PyRosetta free energies of folding positively correlate with the experimentally determined values for these mutants: more positive experimental values are matched by more positive computational predictions, while more negative experimental values are matched by more negative computational predictions (see S1 Fig in S1 File for purely FoldX predictions, which parallel the MD +FoldX results). Indeed, as can be determined by counting the number of mutants in the shaded regions, MD+FoldX correctly predicts the signs of 14 out of 15 mutants, while PyRosetta does so for 11 out of 15 mutants. In general, both MD+FoldX and PyRosetta predict the majority of the mutants to lie in the same relative places on these plots (see S2 Fig in S1 File for a direct comparison of MD+FoldX and PyRosetta predictions). It is moreover pleasing to see

that PyRosetta predicts S70G, which is known to be stabilizing and is thus in some sense a control, to have a negative $\Delta\Delta G_{fold}$ value; MD+FoldX fortunately also yields a reasonably accurate, although not fully stabilizing, prediction for this mutant. In concurrence with the results for free energy distributions presented in the next section, even this relatively small data set demonstrates that the majority of mutants destabilize folding. The $\Delta G_{fold}$ value for $\beta$-lactamase that we determined via experiment is -8.4 kcal/mol. As mutants lose their folding stability as their $\Delta G_{fold}$ values approach 0 and $\Delta G_{fold,mutant} = \Delta G_{fold,wildtype} + \Delta\Delta G_{fold,mutant}$, many of the mutations we have studied that have $\Delta\Delta G_{fold}$ values nearing 8 kcal/mol lie on the verge of unfolding the protein.

Despite the qualitative agreement between the Fig 3 panels, however, they do differ quantitatively. First and foremost, the range of PyRosetta folding free energies is significantly larger than the range of MD+FoldX folding free energies. Much of this difference in range may be attributed to PyRosetta's strongly negative $\Delta\Delta G_{fold}$ values for the S70G, E212K, and K234Q mutants. Without these mutants, PyRosetta's ability to predict the experimental data would significantly decline. MD+FoldX also seems less able to discern experimentally stable from unstable mutants, as it predicts many mutants to be less stable than they are in reality. In combination, these factors contribute to PyRosetta being more strongly correlated with the experimental data, as evidenced by its 0.44 coefficient of determination relative to MD+FoldX's 0.071 coefficient. Indeed, PyRosetta's correlation coefficient of 0.67 for $\beta$-lactamase is among the highest PyRosetta correlation coefficients for proteins published in the literature [73, 74]. To complement our regression analysis, we additionally computed Spearman's rank correlation coefficients [75] for our Experiment vs. MD+FoldX and Experiment vs. PyRosetta data sets. Spearman's rank correlation coefficients assess how correlated the rank order, here based upon $\Delta\Delta G_{fold}$ magnitudes, is between two lists. In concurrence with our regression results, we obtain a rank correlation coefficient of 0.22 for MD+FoldX and 0.44 for PyRosetta, which corroborates the fact that PyRosetta more accurately captures the experimental data, even beyond a simple linear model. Overall, these results suggest that, while current computational tools are not perfect, they can qualitatively predict single mutant free energy trends.

PyRosetta likely outperforms FoldX at quantifying the folding free energies of the Wylie single mutants because the majority of these mutants are solvent accessible (see S5 Fig in S1 File). Past work comparing the accuracies of PyRosetta and FoldX on a combination of Guerois [47] and ProTherm database mutants [76] demonstrated that Rosetta performs best, while FoldX performs worst on mutants involving solvent exposed residues compared with other classes of mutants [57, 74]. This is because FoldX often implausibly favors placing hydrophobic residues on protein surfaces.

**Accuracy of double mutant predictions.** Given their ability to predict folding free energy trends of single residue mutants, for which they were largely designed, we next explored how well these computational techniques performed on double mutants constructed from Wiley data set constituent single mutants. While many multiply-mutated proteins are known to possess free energies of folding that are simply the sums of their constituent single mutation free energies because their constituent mutations act largely independently, some of the most biophysically intriguing mutations possess non-additive free energies and thus lead to epistatic effects that can dictate the course of protein evolution [77–79]. The right-most panel of Fig 4, which plots the folding free energies directly measured for double mutant structures against those obtained by adding the free energies of the constituent single mutants, illustrates that three of the Wiley double mutants, K234Q/R241H, A172P/G283C, and E212K/G218V, possess essentially additive folding free energies, while three others, the A213G/L57H, G144E/L199F,

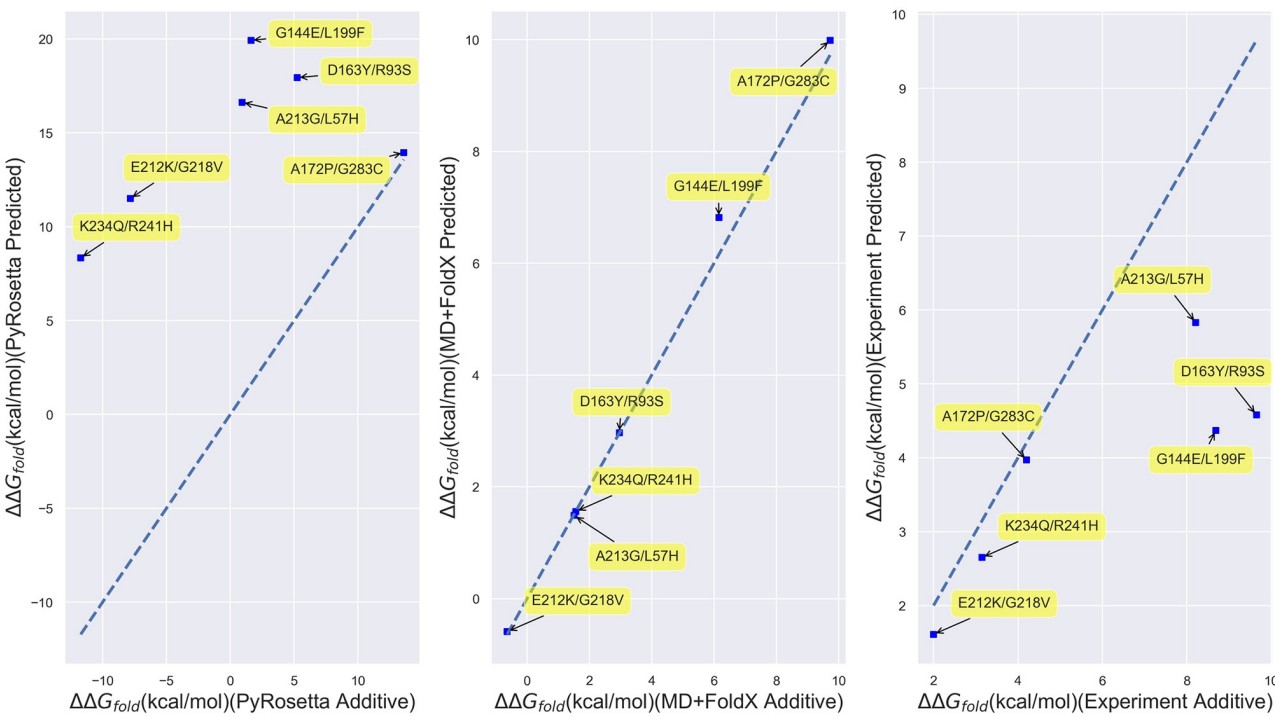

**Fig 4. Scatterplots of directly computed or measured Wylie double mutant free energies vs. the free energies obtained by adding their corresponding constituent single mutant free energies.** **(Left)** Results for PyRosetta; **(Middle)** MD+FoldX; and **(Right)** Experiment. The labels on the points indicate the double mutants involved. The dotted lines are $y = x$ lines, which indicate where computed or measured double mutant free energies are perfectly additive. Those points closest to the plotted lines are thus closest to being additive.

and D163Y/R93S mutants, possess non-additive free energies according to experiment (see S4 Fig in S1 File for a tabulation of the underlying quantitative data). Despite being limited, this set of mutants is thus ripe for benchmarking how predictive computational techniques are for multiply-mutated proteins. Interestingly, we find that, regardless of whether experiment predicts the mutants to be additive or non-additive, FoldX and MD+FoldX always yield additive predictions (middle panel of Fig 4). The additivity of MD+FoldX, even when supplemented with MD relaxation of the original wild type structure, may be anticipated based upon the fact that it does not globally relax mutant conformations. In contrast, PyRo-setta generally yields non-additive predictions (left-most panel of Fig 4). It is because of this non-additivity that PyRosetta outperforms FoldX in predicting the folding free energies of the Wylie double mutants, as depicted in Fig 5. Nevertheless, the fact that PyRosetta's double mutant free energy predictions are always superadditive, likely because it is unable to fully relax double mutant structures, also makes its predictions questionable.

This said, it is noteworthy that both PyRosetta and MD+FoldX are more accurate at predicting the folding free energies of this set of double mutants than the single mutants presented above (see S3 Fig in S1 File for a scatterplot of all of the Wylie mutants). This is likely an artifact of the small double mutant sample size, but it is disconcerting that MD+FoldX achieves this larger double mutant correlation based upon the incorrect assumption of additive free energies and that PyRosetta does so based upon consistently superadditive predictions. All in all, these results call for the development of improved fast, yet accurate techniques that pair a computationally expedient amount of relaxation with empirical, if not optimally physical, functions also parameterized to account for multiply-mutated proteins.

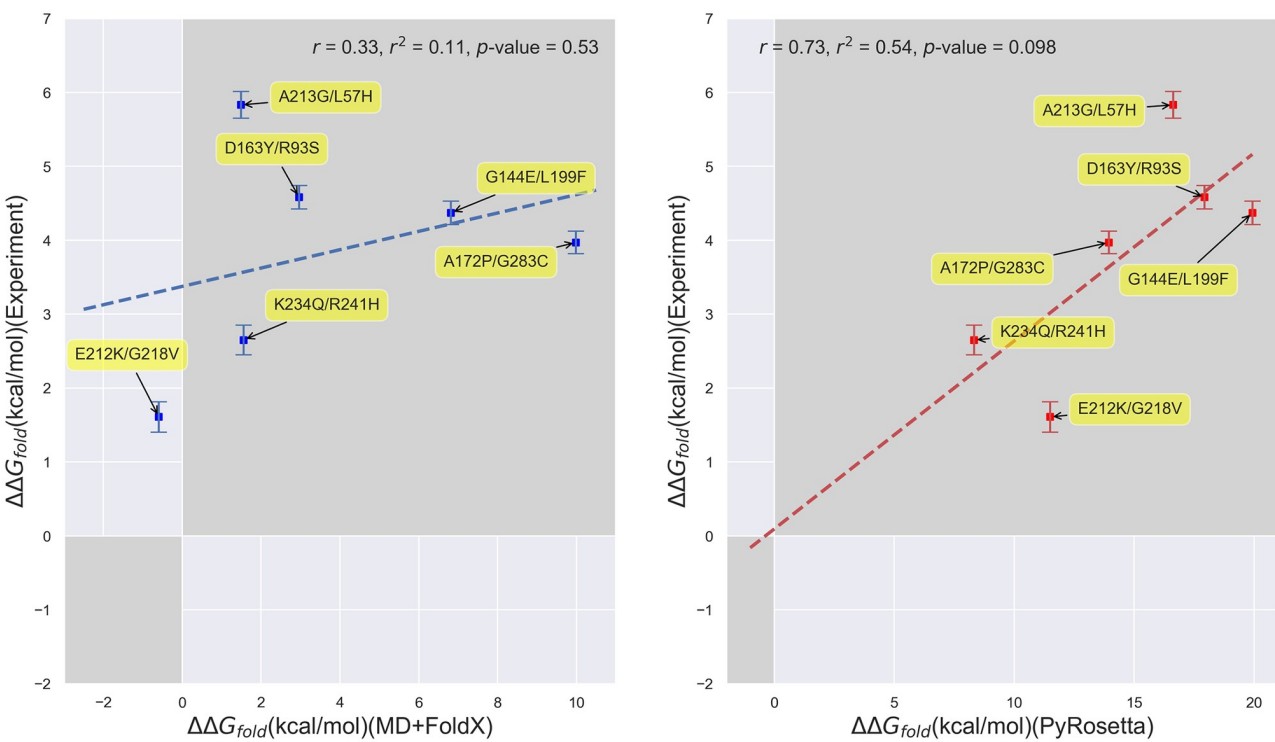

**Fig 5. Scatterplots of experimental vs. predicted folding free energies for the Wylie double mutants. (Left)** Experiment vs. MD+FoldX predictions; **(Right)** Experiment vs. PyRosetta predictions. The shaded regions delineate the first and third quadrants of the plot, which contain mutants whose free energies are of the same sign according to both experiment and computation.

## Firnberg mutant dataset: Analyzing the correlation between free energies of folding and fitness

**Folding free energy distributions.** Encouraged by our computational predictions for the Wylie mutant data set, we next utilized simulation to predict the free energy trends of *all* β-lactamase single mutants with the ultimate aim of characterizing their influence on β-lactamase fitness. We find that the shape of β-lactamase's folding free energy distribution according to FoldX may best be fit by a gamma distribution[2]A gamma distribution (denoted as $\Gamma(\alpha, \beta)$) is characterized by its shape parameter $\alpha$, which determines whether the distribution is exponentially-shaped (for $\alpha \leq 1$) or mounded (for $\alpha > 1$; the greater $\alpha$ is above one, the less skewed the distribution is), and its rate parameter $\beta$, which determines how slowly the distribution decays, with distributions with larger values of $\beta$ decaying more slowly than those with smaller values [80]. (see Fig 6), which can capture the right skew of the distribution due to the substantial number of mutants predicted to have $\Delta\Delta G_{fold} > 10$ kcal/mol. Interestingly, we find only very slight differences between the FoldX and MD+FoldX distributions. MD relaxation of the wild type structure simply shifts some of the probability for observing large free energy mutants to observing more low free energy mutants. The fact that the overall form of the distribution is preserved with and without MD suggests that it is most strongly influenced by the FoldX scoring function. Subsequently, we compared this distribution to that obtained by Firnberg *et al.* with PyRosetta as well as with our own PoPMuSiC results. PoPMuSiC is a popular web server for protein stability prediction (see the Supporting Information for further details regarding the PoPMuSiC algorithm). While the shape of the PyRosetta distribution may also be described by a gamma distribution with a slightly larger right skew, the PoPMuSiC

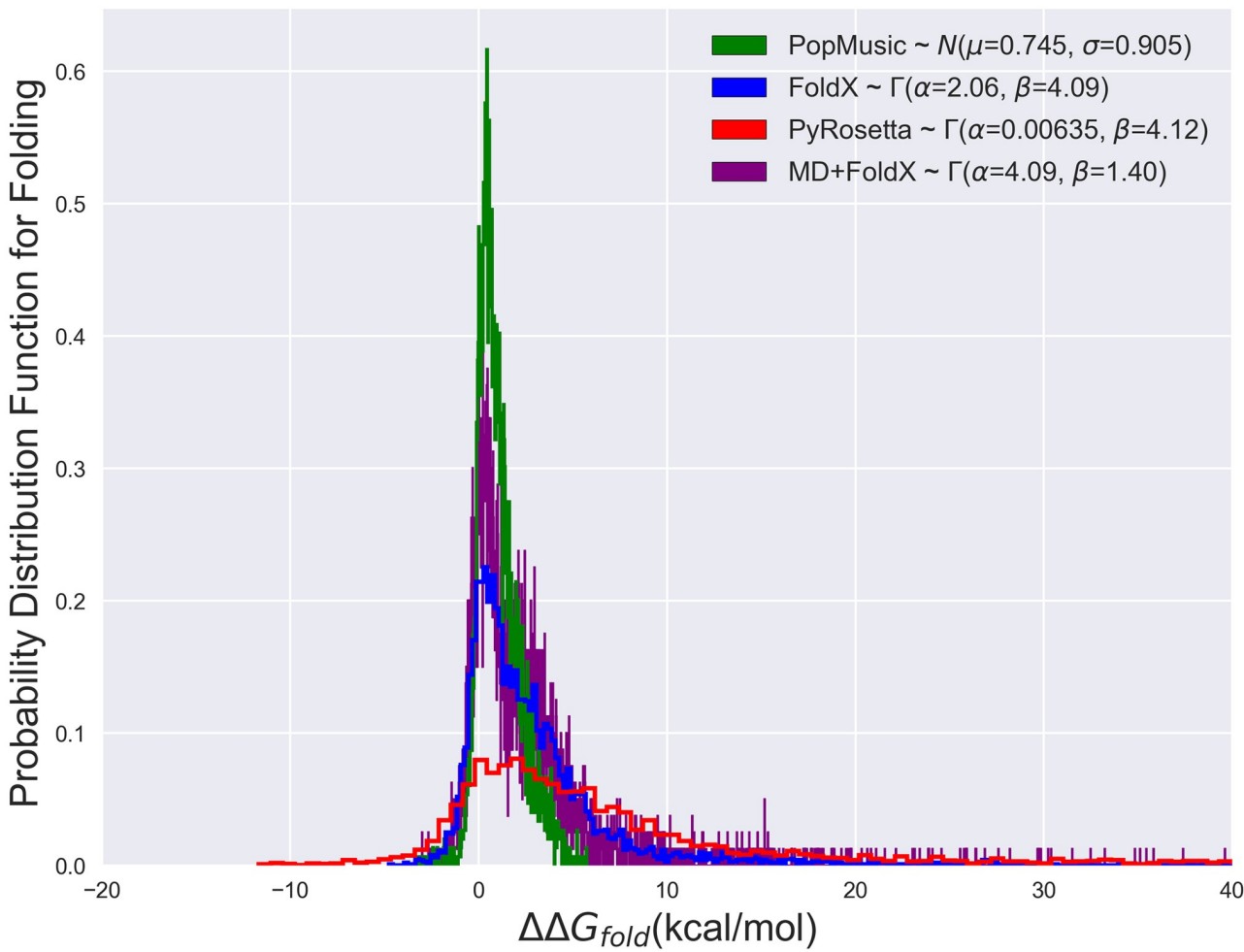

**Fig 6. The probability distribution functions of folding free energies based upon PoPMuSiC, FoldX, PyRosetta, and MD+FoldX results.** The curves depicted are histograms of the data, while the parameters given in the legend are based off of smooth fits. The PoPMuSiC distribution is significantly more peaked and less skewed than the other distributions, making it most consistent with a Gaussian distribution. FoldX, MD+FoldX, and PyRosetta all possess more right-skewed distributions with significant high free energy tails such that their distributions are best captured by $\Gamma$-distributions [80].

distribution was best described by a normal distribution. This is because PoPMuSiC neither considers mutations that destabilize the structure by more than 5 kcal/mol nor those involving a proline, which are likely to induce significant structural modifications [55]. Many of the mutants predicted by FoldX and PyRosetta to be accompanied by large free energy changes stem from these expedient methods' inability to fully relax the structures of mutants in which tryptophans or other volumetrically bulky residues replace volumetrically smaller amino acids (for an illustration, see Fig 8). While experiments find that, in certain cases, tryptophans do in fact grossly destabilize the enzyme [59], in other cases, FoldX and PyRosetta strongly overestimate tryptophan-induced clashes and their related free energies. The key point that may be garnered from this comparison of distributions is that, while disparate computationally-expedient techniques may differ in their predictions for specific mutant free energies, they yield similar free energy trends overall, particularly for mildly destabilizing mutants.

As a further physical check on MD+FoldX's predictions, we additionally analyze its predicted folding free energies as a function of residue number. $\beta$-lactamase crystal structures

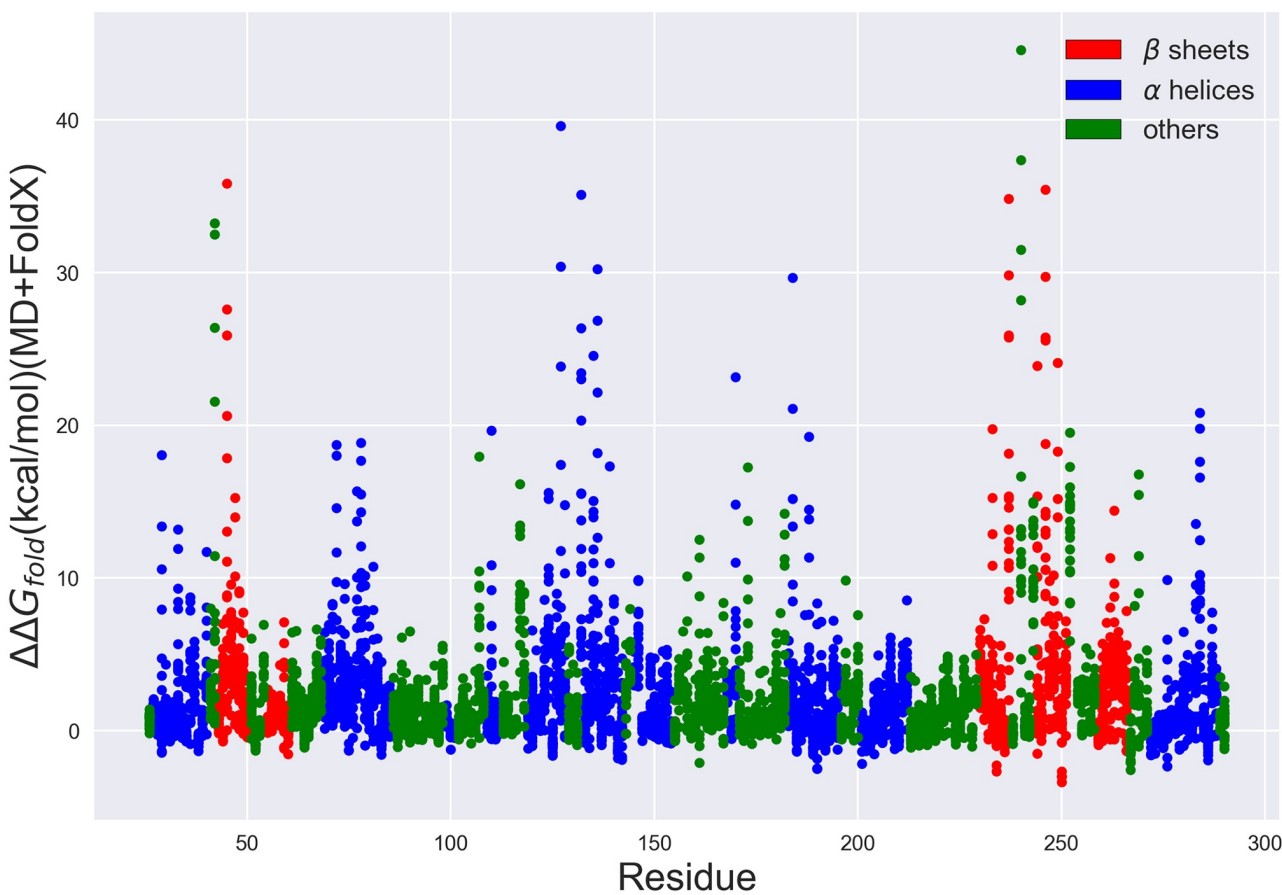

**Fig 7. $\beta$-lactamase folding free energies predicted using MD+FoldX vs. residue number.** Each point depicts a possible single-residue mutant. Those residues located in $\beta$ sheets are depicted in red, those in $\alpha$ helices are depicted in blue, and all others are depicted in green. Many of the mutants with the largest $\Delta\Delta G_{fold}$ values are present in $\beta$ sheets, $\alpha$ helices, or interestingly, around key catalytic residues (such as S70, S130, and K234 as discussed in the Supplemental Information).

reveal that $\beta$-lactamase is composed of two domains, comprised of a total of 5 $\beta$-sheets and 9 $\alpha$-helices [60, 81]. Mutations that disrupt how well these secondary structures form are thus most likely to significantly alter $\beta$-lactamase's folding free energy. In Fig 7, we depict the folding free energies as a function of mutant residue number and color the residues according to the secondary structures they form. It is evident from this plot that many of the largest predicted folding free energy changes occur in regions with stabilizing $\beta$-sheet or $\alpha$-helical character. Since previous work has shown that FoldX predicts $\Delta\Delta G_{fold}$ values essentially as accurately for helix and sheet regions as for all other residues [57], the large free energy changes observed in these regions are likely due to the disruption of secondary structure and therefore serve to validate the predictive capacity of this method.

**Correlation between firnberg fitness data and folding free energies.** With these modeling considerations in mind, we then returned to our original goal of understanding how predictive free energies of folding are of protein fitness by comparing our folding free energies against Firnberg *et al.'s* [38] fitness data. This said, from the left-hand panel of Fig 8 and S2 Table in S1 File, it is clear that folding free energies are reasonable predictors of fitness: many (2175) mutants predicted to be stable with $\Delta\Delta G_{fold} < 5$ kcal/mol are in fact fit, possessing fitness values greater than 0.5 (so-called 'true positives'). We note that the $\Delta\Delta G_{fold}$ and fitness

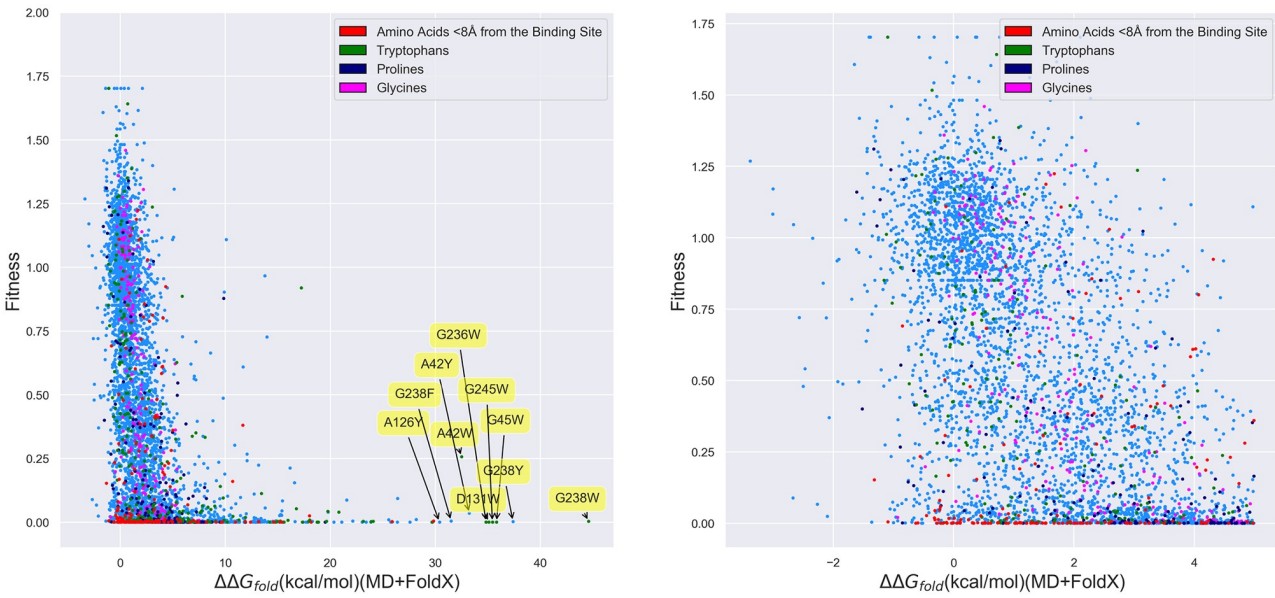

**Fig 8. $\beta$-lactamase fitness values as measured by Firnberg *et al*. vs. MD+FoldX predictions of the folding free energy for all $\beta$-lactamase single mutants. (Right)** All mutants; **(Left)** Mutants with $\Delta\Delta G_{fold}$ < 5 kcal/mol. Mutants involving residues within 8 Å of the binding site and substitutions to tryptophan, proline, and glycine are colored in red, green, purple, and pink, respectively. All other mutants are depicted in blue. As labeled, many of the mutants with the largest predicted free energies of folding involve substitutions to tryptophan (W), resulting in large $\Delta G_{clash}$ terms within FoldX.

cutoffs used in these definitions are somewhat arbitrary, but a $\Delta\Delta G_{fold}$ of 5 kcal/mol was selected for the folding free energy because at 8 kcal/mol, beta-lactamase unfolds and thus, above 5 kcal/mol, it is expected to be unstable. There are additionally many (506) mutants predicted to be unstable, with $\Delta\Delta G_{fold}$ > 5 kcal/mol, that are unfit, possessing fitness values less than 0.5 (so-called 'true negatives'), also as one would hope. As can be inferred from the labeled residues toward the right of the left-hand panel of Fig 8, most of these large free energy mutants involve substitutions of volumetrically smaller residues, such as glycine and alanine, with larger, bulky residues, such as tryptophan and tyrosine, which dramatically raise the free energy contributions associated with steric clash. Remarkably, our plots manifest strikingly few (22) cases for which mutants with large folding free energies (>5 kcal/mol) possess high fitness (>0.5), which we term false negatives. Even though this is heartening, there nevertheless exist numerous (2079) false positives: mutants that exist in the lower left corner of the plot whose small folding free energy differences (<5 kcal/mol), which one would expect to correspond to high fitness values (>0.5), nonetheless map to low fitness values (<0.5). It is also clear from the right-hand panel of Fig 8 that small (< 5 kcal/mol) changes in folding free energies which destabilize the protein, but do not unfold it (based upon its $\Delta G_{fold} \sim -8$ kcal/mol), lead to a wide range of fitness values and are therefore not strongly correlated with fitness. Putting these factors together, we find that roughly 22% of the variance in $\beta$-lactamase fitness can be explained by linearly fitting a regression line to the MD+FoldX data points with $-5 \leq \Delta\Delta G_{fold} \leq 10$ kcal/mol (to remove the influence of large free energy outliers on a linear fit). Better fits that account for mutants with larger folding free energies may be obtained by fitting non-linear functions to the data. As we discuss in the Supplemental Information, the full data set may best be modeled by the offset Duncan Equation, $f(x)_{fold} = \frac{-0.804}{[1+e^{-(x+8.30)/1.17}]^{3597}} + 0.871$

with a correlation coefficient of $r$ = 0.647, an improved correlation coefficient, but still one that struggles to accommodate for the problematic false positives.

To better understand the origin of these many false positives, we have colored mutations within 8 Å of the active site and those involving tryptophan, proline, or glycine in Fig 8. It is well-known that FoldX predictions for mutants containing proline and glycine are often incorrect, as they both disrupt protein secondary structure and glycine side chains are so minimal that replacing them with most other amino acids will result in prohibitive levels of steric clash [57]. As already discussed and clearly indicated by the labeled residues in the figure, substitutions to tryptophans often generate aberrantly large folding free energies. Moreover, mutants located near the active site are more likely to significantly contribute to changes in fitness by affecting enzyme catalysis than protein folding. This has been borne out in one previous study in which the predictive capacity of FoldX for $\beta$-lactamase MIC values was increased from 0.15 to 0.19 by excluding active site residues from consideration [39]. Discarding all of these different mutant classes removes roughly 12% of the false positives, here defined to be mutants having a fitness between 0 and 0.5 and a $\Delta\Delta G_{fold}$ value between -5 and 10 kcal/mol. This boosts the percent of variance in fitness explained by our MD+FoldX calculations based upon linear fitting to 23.8%, a slight, but not profound, improvement over our previous fit.

**Correlation between firnberg fitness and experimental folding free energies.** Given the inability of computational folding free energies to fully explain fitness, one may ask if our lack of predictive power stems from modeling errors. To address this point, in Fig 9, we plot fitness vs. our experimental Wylie free energies. Keeping in mind the limitations of this small data set, we see that there appears to be virtually no correlation between experimental folding free energies and fitness over this small range of folding free energy values. $\Delta\Delta G_{fold}$ values less than 5 kcal/mol may, again, simply not be large enough to induce the structural changes needed to clearly impact fitness. The fact that the same conclusion may be drawn based upon experimental and computational data adds credence to our computational results. While a larger experimental free energy data set may end up manifesting a stronger correlation between fitness and folding free energies, these results lead one to wonder whether taking other potential thermodynamic predictors, such as binding free energy changes, into account may improve matters.

## Improving fitness predictions with binding calculations

Since many of the mutants whose fitness values cannot be well explained by their $\Delta\Delta G_{fold}$ values reside near the active site, going beyond all previous works, we lastly considered these mutants' computational free energies of binding, $\Delta\Delta G_{bind}$. Based upon our ADV docking calculations, we indeed find that many of the mutations that occur within 8 Å of the alpha carbon of S70 significantly increase their mutants' binding free energies. Plotting these 8 Å mutants' fitness against both their folding and binding free energies as in Fig 10 shows that a larger fraction of $\beta$-lactamase fitness can be explained by a combination of $\Delta\Delta G_{fold}$ and $\Delta\Delta G_{bind}$ data. Based upon non-linear fits to the functional forms given in Fig 10 (see the Supporting Information for fitting details), the fit $r$-values increase from 0.244 for folding alone and 0.304 for binding alone to an $r$-value of 0.360 when both folding and binding are accounted for. The larger $r$-value for binding than folding furthermore supports our assumptions about the more important influence of binding on fitness for residues neighboring the active site.

Interestingly, we find that, given $\beta$-lactamase's structure, mutants that affect folding are overwhelmingly independent of mutants that affect binding, as can be seen from the limited number of mutants that cluster around the $\Delta\Delta G_{fold}$-$\Delta\Delta G_{bind}$ diagonal in Fig 10 (we leave the correlation between folding and binding suggested in Fig 7 for the residues neighboring S70, S130, and K234 to future work). We would expect this situation to vary for proteins whose folding and binding mechanisms are more intimately intertwined or that have a more deeply concealed active site.

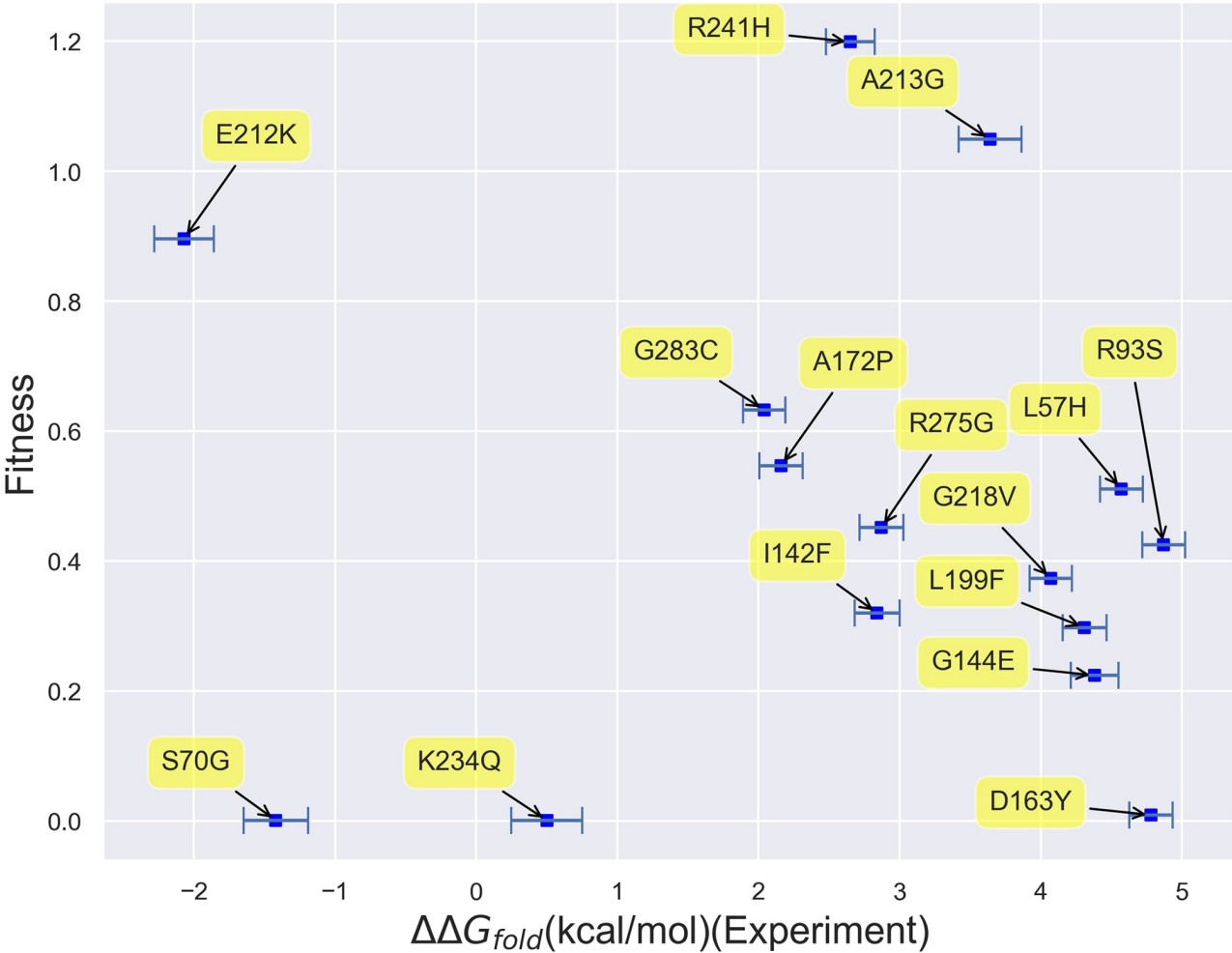

**Fig 9. Scatterplot of fitness vs. Wylie data set experimental free energies of folding.** Virtually no correlation can be observed between fitness and folding free energies based on this data set alone.

As with folding, we can also analyze the form of the fitness vs. free energy of binding curve. As shown in Fig 11, the probability distribution associated with the change in free energies of binding also exhibits a precipitous decline beyond a $\Delta\Delta G_{bind}$ value of roughly 0.25 kcal/mol, a comparatively tight threshold. Thus, the binding distribution may also be characterized by a gamma distribution with a significant right skew. Although we have computed binding free energies for a small set of mutants and a larger set may manifest different trends, we furthermore find that a comparatively small fraction seem to exhibit negative $\Delta\Delta G_{bind}$ values. S8 Fig in S1 File, which labels the largest $\Delta\Delta G_{bind}$ points in Fig 11 with their corresponding mutants, additionally demonstrates that the majority of the mutants that most affect binding alter the critical catalytic residues 70 and 130, as well as several of the residues known to influence catalysis in the 230 range.

## Conclusion

In closing, in this work, we have analyzed how predictive thermodynamic biophysical indicators can be of organismal fitness, focusing in particular on how well protein folding and binding free energies can predict the fitness of $\beta$-lactamase mutants. As a prelude to our fitness

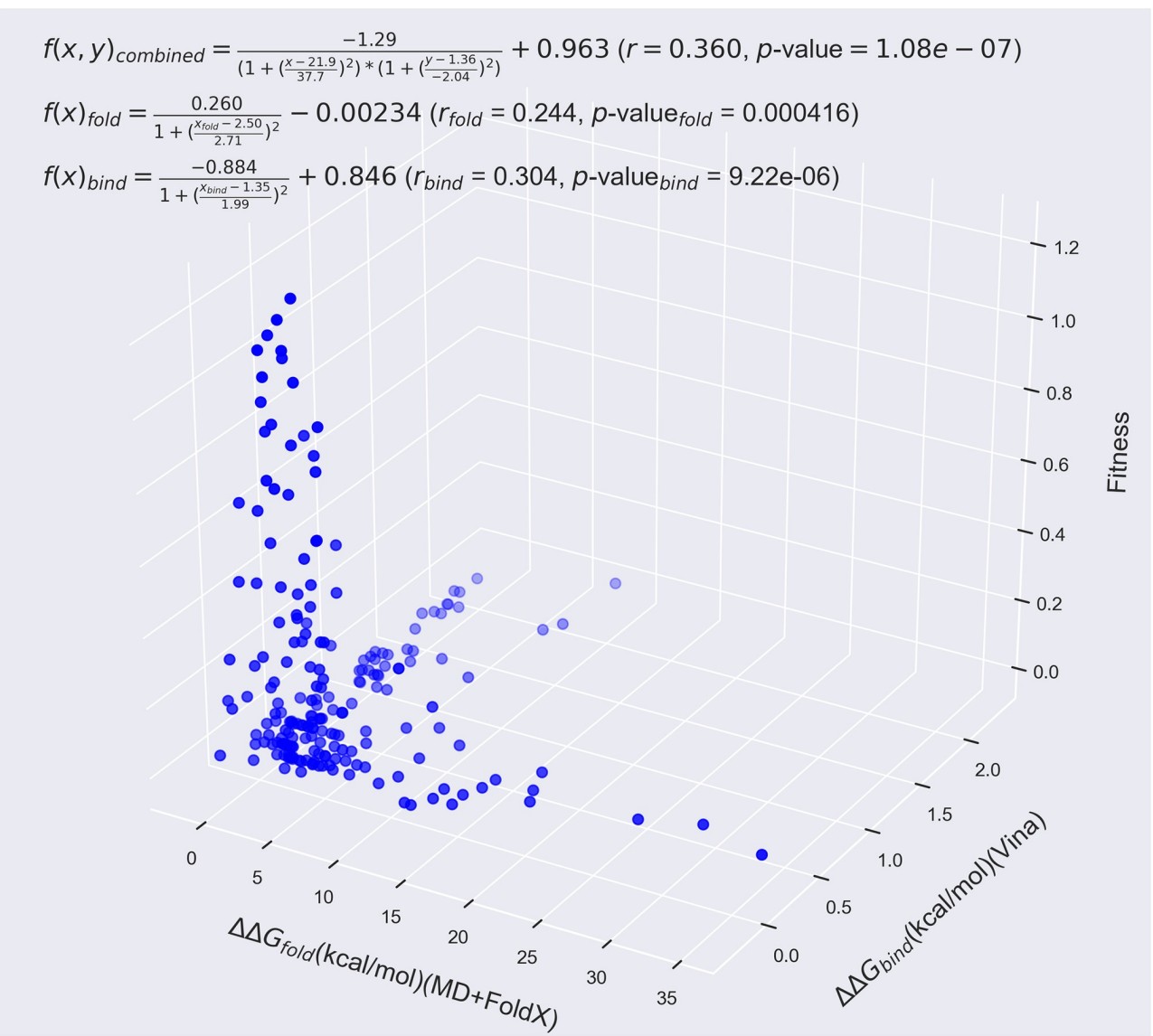

$$f(x, y)_{combined} = \frac{-1.29}{(1 + (\frac{x - 21.9}{37.7})^2) * (1 + (\frac{y - 1.36}{-2.04})^2)} + 0.963 \; (r = 0.360, \; p\text{-value} = 1.08e - 07)$$

$$f(x)_{fold} = \frac{0.260}{1 + (\frac{x_{fold} - 2.50}{2.71})^2} - 0.00234 \; (r_{fold} = 0.244, \; p\text{-value}_{fold} = 0.000416)$$

$$f(x)_{bind} = \frac{-0.884}{1 + (\frac{x_{bind} - 1.35}{1.99})^2} + 0.846 \; (r_{bind} = 0.304, \; p\text{-value}_{bind} = 9.22e\text{-}06)$$

**Fig 10. $\beta$-lactamase fitness values as measured by Firnberg *et al*. vs. MD+FoldX predictions of the folding free energy and Autodock Vina predictions of the binding free energy for residues within 8 Å of the active site.** Using a combination of folding and binding free energies as predictors of fitness significantly improves their predictive capability beyond using them individually by accounting for both of the largely independent (as may be gleaned from the plot) effects of folding and binding. Indeed, the *r*-value of 0.244 of the folding data alone and the *r*-value of 0.304 of the binding data alone are improved to 0.360 when utilizing both data sets to explain the fitness. This data was fit by the two- and three-dimensional non-linear functions provided on the plot.

studies, we first presented the largest published data set of experimental $\beta$-lactamase $\Delta\Delta G_{fold}$ values for mutants purposefully selected to predominantly affect folding. We subsequently demonstrated that trends in these values can be reasonably predicted using high-throughput modeling techniques such as MD+FoldX and PyRosetta. More specifically, we find that while FoldX and PyRosetta can both qualitatively match experimental results, PyRosetta with its more robust conformational sampling algorithms can more quantitatively predict the folding free energies of surface-exposed $\beta$-lactamase single mutants. Interestingly, we find that both MD+FoldX and PyRosetta are capable of making sensible predictions of double mutant free energies of folding, even though they are not explicitly designed to do so and often do so for

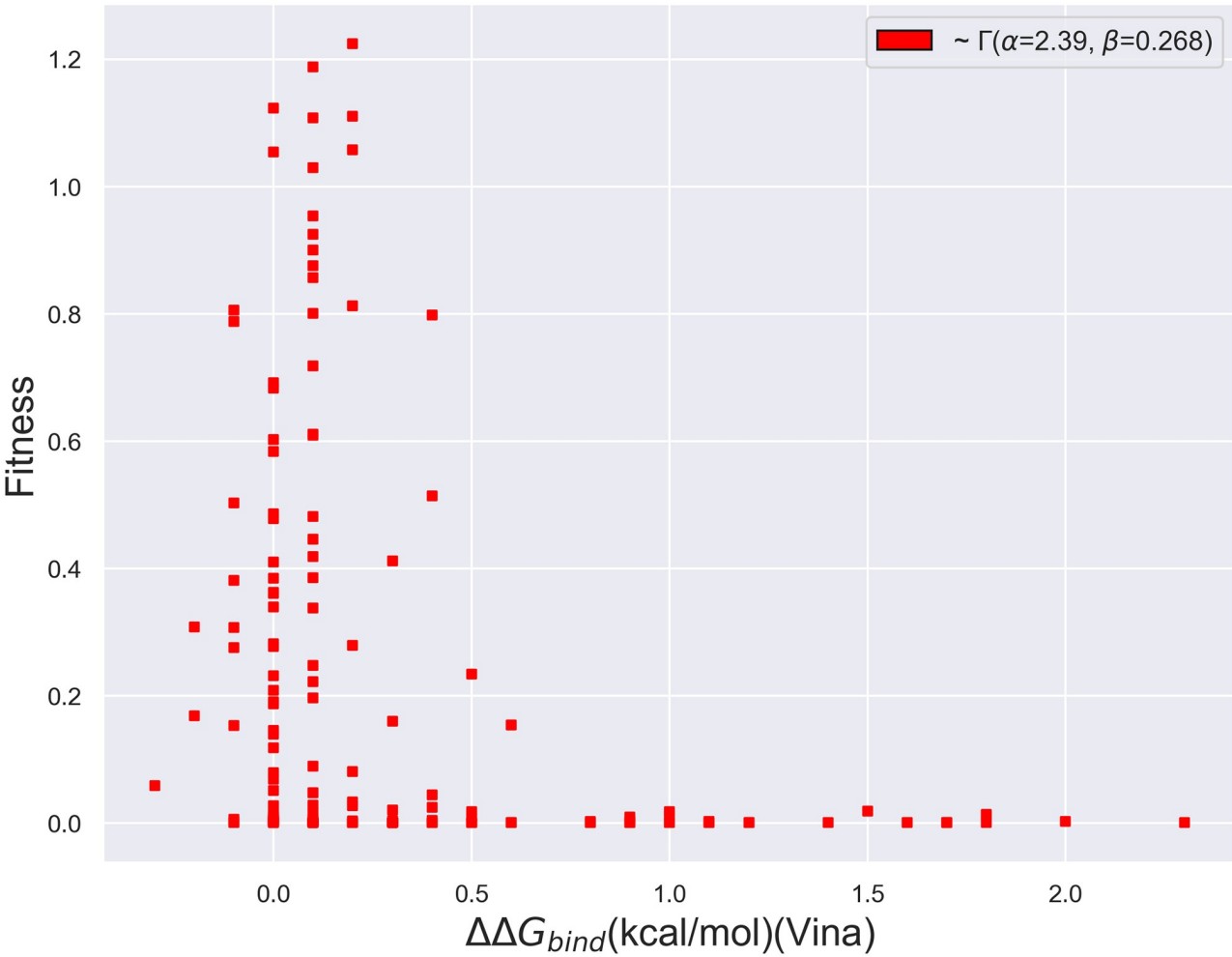

**Fig 11. Scatterplot of the fitness vs. binding free energies produced using AutoDock Vina of $\beta$-lactamase mutants whose mutated residues are within 8 Å of the active site.** The distribution is best captured by a $\Gamma$ distribution with $\alpha = 2.39$ and $\beta = 0.268$.

the wrong physical reasons. Using MD+FoldX predictions and previously acquired $\beta$-lactamase fitness data, we moreover demonstrated that large, positive $\Delta\Delta G_{fold}$ values are highly predictive of low fitness, but that $\Delta\Delta G_{fold}$ values only account for, at most, 24% of the variance in $\beta$-lactamase fitness based on linear models. Adding credence to our simulation results, this low overall predictive capacity was also borne out by comparisons among fitness and experimental folding free energies. Lastly, going beyond previous work, we demonstrated that, for a select set of mutants, the fraction of the fitness that can be accounted for by thermodynamic measures can be improved by including binding free energy information. Nevertheless, the fact that all combinations of our thermodynamic indicators consistently predict a small fraction of the variance in $\beta$-lactamase fitness points to the fact that, to achieve its ambition of predicting fitness landscapes, the community must redouble its efforts to develop and analyze the predictive capacity of other potential predictors of organismal fitness. Even though it is likely that the techniques used here fail to correctly capture some fraction of the variance in the fitness due to their inherent approximations, these techniques have been shown to perform as well as many of the best empirical effective free energy function methods available and thus our results point more to the deficiencies of thermodynamic predictors than to the deficiencies

in our modeling. Indeed, our results strongly suggest that, at least for $\beta$-lactamase, non-thermodynamic measures play a central role in determining fitness.

Although the community's understanding is still evolving, recent research points to the significant impact the kinetics of catalysis [50], protein quality control [10], protein aggregation, degradation, and interactions with other proteins more generally [6], and post-translational modifications [82], among other non-thermodynamic factors, have on fitness. Accurately modeling protein kinetics still represents a formidable challenge for simulators as doing so either requires the ability to simulate out to long enough times to capture all relevant protein dynamics or models that can reliably project out to these long times [83]. Despite the modeling challenges at hand, it may be worthwhile to explore how much of even the short-time dynamics of proteins can predict their fitness, as dynamical fingerprints of proteins have recently been used to uncover new inhibitors [84] and understand allostery [85]. In fact, recent molecular dynamics simulations and NMR experiments performed on a select set of proteins and their mutants have shown that mutations can have "propagatory effects" that can influence the conformation and dynamics of residues up to 25 Å away from them [86–88]. It would be fascinating and worthwhile to eventually be able to relate such propagatory effects to fitness landscapes. Relatedly, seeing which aspects of recent, cutting-edge kinetic models of $\beta$-lactamase [89] can be used to predict fitness would be a next intriguing step. Sufficient progress has also been made toward modeling post-translational modifications that one can readily imagine the incorporation of these effects into high-throughput computational methodologies for mutants in the very near future [90]. The computational modeling of protein quality control and transcriptional and translational dynamics, however, remain in their infancy owing to the difficulty of experimentally determining the strengths and frequencies of the protein-protein and protein-nucleic acid interactions involved which can be used to parameterize reaction network models [91]. Given the wealth of information these models could yield, these represent an exciting frontier that needs significantly more researcher attention moving forward.

Shy of achieving these blue sky ambitions, our results strikingly exemplify the nearer-term need for computational techniques capable of predicting correlations among mutants, which we will loosely term epistatic effects here. Although we studied a limited set of double mutants and their constituent singles using a select set of computationally-expedient techniques, the methods explored tended to yield sensible fits to double mutant data but for the wrong reasons: MD+FoldX implicitly assumed that all double mutants possessed additive free energies, whereas PyRosetta calculated all double mutants to possess superadditive free energies, regardless of whether the mutant experimental free energies were additive or not. Based on previous literature, it is likely that other empirical effective free energy techniques will perform similarly and that many techniques that more completely sample each mutant's conformation space will be too costly to bring to bear on the results of saturation mutagenesis experiments. This glaringly limits our understanding of epistasis to the few, painstakingly obtained complete sets of mutants available, making it difficult, if not impossible, to enunciate just how prevalent epistatic effects are and what ultimate impact they have on evolution across species. While it is true that the FoldX and PyRosetta free energy functions are parameterized on data sets overwhelmingly comprised of single mutants and therefore can certainly be improved via machine learning or better fitting to predict the folding free energies of proteins containing multiple mutations, our data suggest that most of the inadequacies of these techniques stem from their inability to fully relax mutant conformations. All-atom molecular dynamics or Monte Carlo techniques are designed to realize such full relaxation, but typically at costs prohibitive for the high-throughput studies of mutants necessary for understanding fitness landscapes. Thought must therefore be dedicated to how best to relax the regions directly surrounding and connecting mutations while maintaining efficiency. Possible paths to achieving the relaxation needed

may include using umbrella sampling [92], Hamiltonian-exchange-like techniques [93], or simply resolving relaxation protocols compatible with FoldX and PyRosetta that can reliably be used to relax the majority of multiply-mutated proteins.

Even though our results are limited to the $\beta$-lactamase protein, a protein exceptional in that its catalytic function may be directly tied to organismal fitness in bacteria, there is significant evidence that our findings are generalizable across many protein families [23]. We therefore hope that our results motivate the community to develop the beyond-free energy computational tools that will be central to once-and-for-all seizing the holy grail of rapidly and accurately predicting organismal fitness from molecular principles.

## Supporting information

**S1 File. Supplemental information file of supporting discussion and figures.** Contains all supporting discussion regarding $\beta$-lactamase and methods, as well as two supporting tables and nine supporting figures.
(ZIP)

## Acknowledgments

The authors deeply thank Craig Miller, Holly Wichman, Christopher Marx, and many other members of the University of Idaho Center for Modeling Complex Interactions Molecular Modeling working group as well as Hersh Gupta for numerous fruitful discussions. We thank Gabriel Monteiro da Silva for assistance with our binding calculations. We also thank Elad Firnberg for graciously providing us with his fitness data set and related conversations. Computer resources were provided by the Brown Center for Computation and Visualization (CCV) and the high-performance computing center at Idaho National Laboratory, which is supported by the Office of Nuclear Energy of the U.S. DOE and the Nuclear Science User Facilities under Contract No. DE-AC07-05ID14517.

## Author Contributions

**Conceptualization:** Jagdish Suresh Patel, Daniel M. Weinreich, Brenda M. Rubenstein.

**Data curation:** Jordan Yang, Nandita Naik, Christopher S. Wylie, Wenze Gu, Jessie Huang, Mandar T. Naik.

**Formal analysis:** Jordan Yang, Nandita Naik, Jagdish Suresh Patel, Christopher S. Wylie, Wenze Gu, Jessie Huang, F. Marty Ytreberg, Mandar T. Naik, Brenda M. Rubenstein.

**Funding acquisition:** Jagdish Suresh Patel, F. Marty Ytreberg, Brenda M. Rubenstein.

**Investigation:** Mandar T. Naik, Daniel M. Weinreich, Brenda M. Rubenstein.

**Methodology:** Jordan Yang, Nandita Naik, Christopher S. Wylie, Mandar T. Naik, Brenda M. Rubenstein.

**Project administration:** F. Marty Ytreberg, Daniel M. Weinreich.

**Software:** Jordan Yang, Jagdish Suresh Patel, Jessie Huang.

**Supervision:** Daniel M. Weinreich, Brenda M. Rubenstein.

**Visualization:** Jordan Yang, Wenze Gu.

**Writing – original draft:** Brenda M. Rubenstein.

**Writing – review & editing:** Jordan Yang, Jagdish Suresh Patel, Mandar T. Naik, Brenda M. Rubenstein.

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
