## [Decision Letter · Decision Letter 0]

22 Apr 2020

PONE-D-20-04028

Predicting the Viability of Beta-Lactamase: How Folding and Binding Free Energies Correlate with Beta-Lactamase Fitness

PLOS ONE

Dear Dr. Rubenstein,

Thank you for submitting your manuscript to PLOS ONE. After careful consideration, we feel that it has merit but does not fully meet PLOS ONE’s publication criteria as it currently stands. Therefore, we invite you to submit a revised version of the manuscript that addresses the points raised during the review process.

Your manuscript has been examined by two experts in the field. They both find your work of interest, but point to a number of issues that must be convincingly addressed in a revised version.

We would appreciate receiving your revised manuscript by Jun 06 2020 11:59PM. To enhance the reproducibility of your results, we recommend that if applicable you deposit your laboratory protocols in protocols.io, where a protocol can be assigned its own identifier (DOI) such that it can be cited independently in the future. For instructions see: http://journals.plos.org/plosone/s/submission-guidelines#loc-laboratory-protocols

We look forward to receiving your revised manuscript.

Kind regards,

Jose M. Sanchez-Ruiz

Academic Editor

PLOS ONE

Reviewers' comments:

Reviewer's Responses to Questions

**Comments to the Author**

1. Is the manuscript technically sound, and do the data support the conclusions?

Reviewer #1: Yes

Reviewer #2: Yes

2. Has the statistical analysis been performed appropriately and rigorously? 

Reviewer #1: Yes

Reviewer #2: Yes

3. Have the authors made all data underlying the findings in their manuscript fully available?

Reviewer #1: Yes

Reviewer #2: Yes

4. Is the manuscript presented in an intelligible fashion and written in standard English?

Reviewer #1: Yes

Reviewer #2: Yes

5. Review Comments to the Author

Reviewer #1: The authors of this work employ multiple computational servers to predict and rationalise the changes in free energy changes obtained from specific single-point and double mutations on beta-lactamase. They take a step further to explore how well fitness correlates with change in stability measured from experiments. The work is written well

and though the results are not positive I enjoyed reading the overall thought process associated with their discussion and the need for fitness prediction algorithms. I have the following comments on their work which should be addressed.

1) Correlation in itself does not mean a good predictive ability as the mutant free energy changes should also lie in the correct quadrant. The authors should highlight the four quadrants in plots where experimental and predicted stability changes are compared. This makes it easier for the reader to quickly get an estimate of how many are correctly predicted apart from knowing how well they are predicted.

2) Fig. 4: It is tough to say whether PyRosetta or FOLDX performs better here as there are only 6 data points. The number of data points should be a lot more to authoritatively say that one method works better than the other. The correlations in this case therefore make little sense as can also be seen from the p-values.

3) Fig. 7: In the discussion associated with this figure, the authors say that there is 'folding free energies are reasonable predictors of fitness for many residues". How do they come to this conclusion and what fraction of the points 'correlate' well? This number alone could give some insight as all Fig 7 suggests is that DDG cannot be a good measure of fitness.

4) Adding to the point above, DDG need not be the one factor that can affect fitness as quantified here. There are numerous factors associated with a protein including kinetic stability , diffusivity with in a cell , quinary interactions that aid or impede an organisms fitness in a myriad number of ways. These could be one other factors that could confound the observations in Figure 7.

5) On the effect of mutations: it is likely that most mutational servers should also consider propagatory effects which are not currently included. For example, see recent works on this in PMIDs: 30268910, 28910120, 27958720. The authors could consider discussing these as one of the reasons why PyRosetta or FOLDX perform worse when predicting changes in stability.

Reviewer #2: The manuscript by Yang et al. compared the folding free energies of 21 TEM-1 b-lactamase single and double mutants and previous fitness data. The concept of this study is very interesting and within the scope of Plos One. However, some unclear points should be addressed before publication

page 2: differential scanning calorimetry…. any references for the methods/tools would be extremely helpful;

page 2: be determined via isothermal titration calorimetry or surface plasmon resonance…… any references for the methods/tools would be extremely helpful;

Figure 1: add the pdb entry for the TEM-1 structure

Page5:. The thermodynamic stability of each allele was determined by circular dichroism (CD) on a Jasco J-815 spectrometer… What buffer/pH?

Page 5: In methodology, it is stated that the melting temperature and van´t Hoff enthalpy was fitted to a two-state transition, however, none of the figures and result are shown (a table listing all result would be helpful)

Pag 6 “it does not contain information regarding TEM-1 and thus FoldX has neither been explicitly fit nor tested on the TEM-1 mutants analysed here”. …….. I do not think this main conclusion of the study is correct. See PROTEINS 23 63-72 (1995) PMID: 8539251 included in the ProTherm database.

Please change “wildtype” on page 8 to “wild type”

- Figura s2 is not mentioned

- Figura s3 is not mentioned

- Figura s7 is not mentioned

Reference: 35,41,57,69,70,72,77: Remove DOI number please.

Supporting information:

Page 6 : “10^6 cfu/mL, not 106 cfu/mL”

Figure S4. Date of the Wylie mutants are incomplete

I found Figure S6 to be confusing. Some Wylie mutants are missing (A172P, G218V, S70G) and there are others mutants (not named in the manuscript)

6. PLOS authors have the option to publish the peer review history of their article (what does this mean?). If published, this will include your full peer review and any attached files.

Reviewer #1: No

Reviewer #2: No

---

## [Author Response · Author response to Decision Letter 0]

4 May 2020

We uploaded our response letter, however, our responses are copied below. Please refer to our response letter, as it is properly formatted. 

Dear Dr. Heber and Reviewers,

 We graciously thank you and the reviewers for their time reviewing our manuscript. Please find our point-by-point responses below. 

Review Comments to the Authors

Reviewer #1: The authors of this work employ multiple computational servers to predict and rationalise the changes in free energy changes obtained from specific single-point and double mutations on beta-lactamase. They take a step further to explore how well fitness correlates with change in stability measured from experiments. The work is written well and though the results are not positive I enjoyed reading the overall thought process associated with their discussion and the need for fitness prediction algorithms.

First and foremost, we thank the reviewer for his/her/their time carefully reviewing our manuscript and are pleased that the reviewer found it enjoyable to read. We have done our best to address your insightful comments below. All changes to our manuscript are indicated in blue in the manuscript and supplemental information documents 

 I have the following comments on their work which should be addressed.

1) Correlation in itself does not mean a good predictive ability as the mutant free energy changes should also lie in the correct quadrant. The authors should highlight the four quadrants in plots where experimental and predicted stability changes are compared. This makes it easier for the reader to quickly get an estimate of how many are correctly predicted apart from knowing how well they are predicted. 

We thank the reviewer for this suggestion, which will make our results easier to understand. Following your advice, we have shaded the first and third quadrants of main text Figures 2 and 4, and Supplementary Figure 3. By shading these quadrants, we make it clear which points are predicted to be positive by both experimental and computational techniques and which are predicted to be negative by both sets of techniques. This shading complements what we discussed in our original manuscript: 

“As illustrated in Figure 2, which compares experimental and computational free energies for the single mutants, both MD+FoldX and PyRosetta free energies of folding positively correlate with the experimentally determined values for single mutants: more positive experimental values are matched by more positive computational predictions (see Supporting Figure S1 for purely FoldX predictions, which parallel the MD+FoldX results). It is gratifying to see that both methods predict the majority of the mutants to lie in the same relative places on these plots, with mutants that induce negative effects on free energies of folding experimentally inducing negative or near-negative effects on free energies computationally and vice-versa. Overall, MD+FoldX correctly predicts the signs of 14 out of 15 mutants, while PyRosetta does so for 11 out of 15 mutants.” 

However, building upon your suggestions, we have incorporated a discussion of the shading into our new main text description of this figure and clarified our discussion: 

“In Figure 2, we plot experimentally-determined folding free energies against computationally-predicted free energies for the single mutants. We have shaded the first and third quadrants in this figure to ease identification of the mutants whose experimental and computational free energies are of the same sign. It is thus gratifying to see that both MD+FoldX and PyRosetta free energies of folding positively correlate with the experimentally determined values for these mutants: more positive experimental values are matched by more positive computational predictions, while more negative experimental values are matched by more negative computational predictions (see Supporting Figure S1 for purely FoldX predictions, which parallel the MD+FoldX results). Indeed, as can be determined by counting the number of mutants in the shaded regions, MD+FoldX correctly predicts the signs of 14 out of 15 mutants, while PyRosetta does so for 11 out of 15 mutants. In general, both MD+FoldX and PyRosetta predict the majority of the mutants to lie in the same relative places on these plots (see Figure S2 for a direct comparison of MD+FoldX and PyRosetta predictions). It is moreover pleasing to see that PyRosetta predicts S70G, which is known to be stabilizing and is thus in some sense a control, to have a negative ΔΔGfold value; MD+FoldX fortunately also yields a reasonably accurate, although not fully stabilizing, prediction for this mutant.”

We have moreover added the following text to our captions: 

“The shaded regions delineate the first and third quadrants of the plot, which contain mutants whose free energies are of the same sign according to both experiment and computation.”

2) Fig. 4: It is tough to say whether PyRosetta or FOLDX performs better here as there are only 6 data points. The number of data points should be a lot more to authoritatively say that one method works better than the other. The correlations in this case therefore make little sense as can also be seen from the p-values. 

We thank the reviewer for this comment as we whole-heartedly agree. Experiments aimed at creating and measuring the free energies of folding of a wider array of double mutants are underway, although, particularly due to COVID-19, we will not have access to these for quite some time. Thus, we have to make judgements based on the mutants we have. 

This said, in the original manuscript, we indicated our agreement with your statement in the following sentence: 

“Despite being limited, this set of mutants is thus ripe for benchmarking how predictive computational techniques are for multiply-mutated proteins.”

After carefully perusing the manuscript, we also never said that PyRosetta is better than FoldX at predicting mutant free energies based upon the r2 values provided in the figures. Our argument was predicated on the fact that regardless of whether the double mutantss’ folding free energies are sums of their constituent single mutants’ folding free energies or not, FoldX always predicts their free energies to be additive, which is clearly incorrect. PyRosetta at least leaves room for its simulations to yield nonadditive double mutant free energies, even though these are also not particularly accurate. We made this argument in the original manuscript through the following text: 

“Interestingly, we find that, regardless of whether experiment predicts the mutants to be additive or non-additive, FoldX and MD+FoldX always yield additive predictions (middle panel of Figure 3). The additivity of MD+FoldX, even when supplemented with MD relaxation of the original wild type structure, may be anticipated based upon the fact that it does not globally relax mutant conformations. In contrast, PyRosetta generally yields non-additive predictions (left-most panel of Figure 3). It is because of this non-additivity that PyRosetta (r=0.73) outperforms FoldX (r=0.33) in predicting the folding free energies of the Wylie double mutants, as depicted in Figure 4. Nevertheless, the fact that PyRosetta's double mutant free energy predictions are always superadditive, likely because it is unable to fully relax double mutant structures, also makes its predictions questionable.”

To clarify that we are not entirely basing our arguments in the above text on r-values, we have removed the parenthetical references to these values in our revised text. We have nevertheless chosen to keep them on our plot for reference, as they are accompanied by accurate p-values which should help the reader analyze significance of the correlation. 

3) Fig. 7: In the discussion associated with this figure, the authors say that there is 'folding free energies are reasonable predictors of fitness for many residues". How do they come to this conclusion and what fraction of the points 'correlate' well? This number alone could give some insight as all Fig 7 suggests is that DDG cannot be a good measure of fitness. 

We agree with the reviewer that the data in Figure 7 show that free energies do not correlate perfectly with fitness values. In fact, that’s the crux of the paper and we state this in several places in the original manuscript, including in the Introduction: 

“We find that folding free energies account for, at most, 24% of the variance in beta-lactamase fitness values according to linear models and, somewhat surprisingly, complementing folding free energies with computationally-predicted binding free energies of residues near the active site only increases the folding-only figure by a few percent. This strongly suggests that the majority of beta-lactamase's fitness is controlled by factors other than free energies.” 

Our statement that the free energies are reasonable (note that we said reasonable, not good or strong) predictors of fitness is primarily based on the fact that our computational free energy predictions yield relatively few false negatives (22) in comparison with the number of true negatives (506) and positives (2175) they yield. The number of false negatives, false positives, true negatives, and true positives our modeling obtains is presented below and has been added as Figure S4 to the Supplementary Materials: 

We have defined false negatives to be mutants predicted to be unstable (ΔΔG > 5 kcal/mol) yet are experimentally determined to be fit (Fitness > 0.5), false positives to be mutants that are predicted to be stable (ΔΔG > 5 kcal/mol) yet are experimentally unfit (Fitness < 0.5), true negatives to be mutants that are predicted to be unstable (ΔΔG > 5 kcal/mol) and are unfit (Fitness < 0.5), and true positives to be mutants that are predicted to be stable (ΔΔG < 5 kcal/mol) and are fit (Fitness > 0.5). We note that the ΔΔG and fitness cutoffs used in these definitions are somewhat arbitrary, but a ΔΔGfold of 5 kcal/mol was selected for the folding free energy because, at 8 kcal/mol, beta-lactamase unfolds and thus, above 5 kcal/mol, it is expected to be unstable.

In the original manuscript, we indicated that this is what we meant by reasonable predictor in the following text: 

“From the left-hand panel of Figure 7, it is clear that folding free energies are reasonable predictors of fitness for many residues: many mutants with negative or near neutral effects on folding free energies also possess large fitness values. There is additionally an abundance of mutants for which large positive free energy predictions correlate with low fitness, also as one would hope...Remarkably, our plots manifest strikingly few cases for which mutants with large folding free energies (>5 kcal/mol) possess high fitness, which we will term false negatives. Even though this is heartening, there nevertheless exist numerous false positives: mutants that exist in the lower left corner of the plot whose small folding free energy differences, which one would expect to correspond to high fitness values, nonetheless map to low fitness values.”

Based upon your comments, we decided to quantify these statements by including a more thorough discussion of true positives, true negatives, false positives, and false negatives in the text in addition to a discussion of how many of the mutants fall into these four categories: 

“From the left-hand panel of Figure 7 and Supplemental Information Table S4, it is clear that folding free energies are reasonable predictors of fitness: many (2175) mutants predicted to be stable with ΔΔGfold<5 kcal/mol are in fact fit, possessing fitness values greater than 0.5 (so-called 'true positives') \\footnote{We note that the ΔΔGfold and fitness cutoffs used in these definitions are somewhat arbitrary, but a ΔΔGfold of 5 kcal/mol was selected for the folding free energy because at 8 kcal/mol, beta-lactamase unfolds and thus, above 5 kcal/mol, it is expected to be unstable.}. There are additionally many (506) mutants predicted to be unstable, with ΔΔGfold>5 kcal/mol, that are unfit, possessing fitness values less than 0.5 (so-called 'true negatives'), also as one would hope. As can be inferred from the labeled residues toward the right of the left-hand panel of Figure 7, most of these large free energy mutants involve substitutions of volumetrically smaller residues, such as glycine and alanine, with larger, bulky residues, such as tryptophan and tyrosine, which dramatically raise the free energy contributions associated with steric clash. Remarkably, our plots manifest strikingly few (22) cases for which mutants with large folding free energies (> 5 kcal/mol) possess high fitness (> 0.5), which we term false negatives. Even though this is heartening, there nevertheless exist numerous (2079) false positives: mutants that exist in the lower left corner of the plot whose small folding free energy differences (< 5 kcal/mol), which one would expect to correspond to high fitness values (> 0.5), nonetheless map to low fitness values (< 0.5). It is also clear from the right-hand panel of Figure 7 that small (< 5 kcal/mol) changes in folding free energies which destabilize the protein, but do not unfold it (based upon its ΔGfold~ -8 kcal/mol), lead to a wide range of fitness values and are therefore not strongly correlated with fitness.”

4) Adding to the point above, DDG need not be the one factor that can affect fitness as quantified here. There are numerous factors associated with a protein including kinetic stability , diffusivity with in a cell, quinary interactions that aid or impede an organisms fitness in a myriad number of ways. These could be one other factors that could confound the observations in Figure 7. 

Again, we absolutely agree and this is one of the key punchlines of the manuscript. For example, in the Abstract, we note that: 

“This strongly suggests that the majority of beta-lactamase's fitness is controlled by factors other than free energies.”

In the Introduction, we say: 

“Yet, even in those rare instances, even the simplest protein's fitness is influenced by a wide variety of factors including protein and gene expression levels, interactions with chaperones, protein folding stability, protein folding dynamics, and proteolytic susceptibility -- as well as many complex factors yet to be uncovered or understood.”

As well as: 

“Nevertheless, as may be expected given the complexity of the overall transcription, translation, and post-translation processes, we demonstrate that thermodynamic descriptors only explain a small fraction of beta-lactamase fitness results.”

And, in the Conclusion, we state:

“Even though it is likely that the techniques used here fail to correctly capture some fraction of the variance in the fitness due to their inherent approximations, these techniques have been shown to perform as well as many of the best empirical effective free energy function methods available and thus our results point more to the deficiencies of thermodynamic predictors than to the deficiencies in our modeling. Although the community's understanding is still evolving, recent research points to the significant impact the kinetics of catalysis \\cite{Knies_MBE_2017}, protein quality control \\cite{Bershtein_MolCell_2013}, protein aggregation and degradation \\cite{DePristo_NatRevGen_2005}, and post-translational modifications \\cite{Brunk_PNAS}, among other non-thermodynamic factors, have on fitness.” 

To make both of our points clearer, we have added the following sentence to the Conclusion: 

“Indeed, our results strongly suggest that, at least for beta-lactamase, non-thermodynamic measures play a central role in determining fitness.”

You bring up specific examples of non-thermodynamic metrics such as “kinetic stability, diffusivity within a cell, and quinary interactions.” I believe we have addressed kinetic stability and diffusivity when we mentioned “kinetics of catalysis” in the above. To ensure that we address quinary interactions, we have modified the above to explicitly mention protein-protein interactions: 

“Although the community's understanding is still evolving, recent research points to the significant impact the kinetics of catalysis \\cite{Knies_MBE_2017}, protein quality control \\cite{Bershtein_MolCell_2013}, protein aggregation, degradation, and interactions with other proteins more generally \\cite{DePristo_NatRevGen_2005}, and post-translational modifications \\cite{Brunk_PNAS}, among other non-thermodynamic factors, have on fitness.”

5) On the effect of mutations: it is likely that most mutational servers should also consider propagatory effects which are not currently included. For example, see recent works on this in PMIDs: 30268910, 28910120, 27958720. The authors could consider discussing these as one of the reasons why PyRosetta or FOLDX perform worse when predicting changes in stability.

We agree with the reviewer that FoldX and PyRosetta only capture the essentially static and short-range effects of mutations. Our group is in fact working on relating intraprotein residue interaction networks to fitness landscapes, very much as described in the papers suggested, of which we were previously unaware (thank you for suggesting these; they were very helpful given this more recent thrust). That said, it is difficult to simulate all single and higher order mutants of a given protein using molecular dynamics to most accurately capture the propagatory effects you describe and hence we do not address them in this paper.

Based on your recommendations, we have incorporated a discussion of these effects and the suggested papers into our Conclusion: 

“In fact, recent molecular dynamics simulations and NMR experiments performed on a select set of proteins and their mutants have shown that mutations can have ``propagatory effects'' that can influence the conformation and dynamics of residues up to 25 Å away from them \\cite{naganathan2019modulation,rajasekaran2017general,rajasekaran2017universal}. It would be fascinating and worthwhile to eventually be able to relate such propagatory effects to fitness landscapes.”

We thank the reviewer again for carefully scrutinizing our manuscript, and in particular, for his/her/their recommendations of the "propagatory effects" manuscripts.

Reviewer #2: The manuscript by Yang et al. compared the folding free energies of 21 TEM-1 b-lactamase single and double mutants and previous fitness data. The concept of this study is very interesting and within the scope of Plos One. However, some unclear points should be addressed before publication

We thank the reviewer for his/her/their time carefully reviewing our manuscript and for finding our paper interesting. We do our best to address your points below. 

page 2: differential scanning calorimetry…. any references for the methods/tools would be extremely helpful; 

Thank you for pointing out our omission of these references. We have now added the following references about differential scanning calorimetry to our Introduction:

Christopher M. Johnson. Differential scanning calorimetry as a tool for protein folding and stability. Archives of Biochemistry and Biophysics. 531, 100-109 (2013). https://doi.org/10.1016/j.abb.2012.09.008

Ernesto Freire. Differential scanning calorimetry. Protein Stability and Folding. 191-218 (1995). 

https://link.springer.com/protocol/10.1385/0-89603-301-5:191

page 2: be determined via isothermal titration calorimetry or surface plasmon resonance…… any references for the methods/tools would be extremely helpful;

Thank you again for pointing out these omissions. We have now added the following references about isothermal titration calorimetry and surface plasmon resonance to our Introduction:

Isothermal titration calorimetry

Stephanie Leavitt and Ernesto Friere. Direct measurement of protein binding energetics by isothermal titration calorimetry. Current Opinion in Structural Biology. 11 (5): 560-566 (2001). https://doi.org/10.1016/S0959-440X(00)00248-7

Surface plasmon resonance imaging: 

Richard B. M. Schasfoort. Chapter 1: Introduction to Surface Plasmon Resonance in Handbook of Surface Plasmon Resonance (2), 1-26 (2017). https://pubs.rsc.org/en/content/chapterhtml/2017/bk9781782627302-00001?isbn=978-1-78262-730-2&sercode=bk

Figure 1: add the pdb entry for the TEM-1 structure 

As suggested, we have added the pdb reference for TEM-1, 1xpb, to the Figure.

Page5:. The thermodynamic stability of each allele was determined by circular dichroism (CD) on a Jasco J-815 spectrometer… What buffer/pH?

Thank you for pointing this omission out. We performed our circular dichroism experiments in a 200 mM potassium phosphate pH 7.0 with 4% glycerol buffer. We have added this information to the following sentence of the manuscript: 

“The thermodynamic stability of each allele was determined by circular dichroism (CD) on a Jasco J-815 spectrometer in a 200 mM potassium phosphate pH 7.0 with 4% glycerol buffer.”

Page 5: In methodology, it is stated that the melting temperature and van´t Hoff enthalpy was fitted to a two-state transition, however, none of the figures and result are shown (a table listing all result would be helpful) 

Based upon your request, we have added a new table (Table S1) under the section entitled “Wylie Mutant Circular Dichroism Data at 25℃” to the supplement containing all mutants’ melting temperatures, enthalpies, and free energies. 

Pag 6 “it does not contain information regarding TEM-1 and thus FoldX has neither been explicitly fit nor tested on the TEM-1 mutants analysed here”. …….. I do not think this main conclusion of the study is correct. See PROTEINS 23 63-72 (1995) PMID: 8539251 included in the ProTherm database. 

Upon further reviewing the ProTherm database, we found that the reviewer is 100% correct. There are actually many different beta-lactamase entries in the ProTherm database, but only several related to the protein we studied here, TEM-1. However, this point actually strengthens what is the main argument of our study, which is that FoldX, PyRosetta, and free energy changes are not predictive of fitness for the variety of reasons we have articulated. Beta-lactamase’s presence in the ProTherm database emphasizes that FoldX is even less predictive given its training than one would hope. 

To rectify the text, we have deleted this sentence on page 6, instead saying that: 

“Of relevance to this work, FoldX is especially designed to model single mutants and has been trained on the select set of beta-lactamase mutants found in the ProTherm database, but has been infrequently applied to multi-point mutants \\cite{Bershtein_Nature_2006}.”

Please change “wildtype” on page 8 to “wild type” 

We have made this edit, as suggested.

- Figura s2 is not mentioned 

We have now added a reference to this Figure in the following text on page 10: 

“In general, both MD+FoldX and PyRosetta predict the majority of the mutants to lie in the same relative places on these plots (see Figure S2 for a direct comparison of MD+FoldX and PyRosetta predictions).”

- Figura s3 is not mentioned 

We have now added a reference to Figure S3 on page 15 in the following text: 

“This said, it is noteworthy that both PyRosetta and MD+FoldX are more accurate at predicting the folding free energies of this set of double mutants than the single mutants presented above (see Figure S3 for a scatterplot of all of the Wylie mutants).”

- Figura s7 is not mentioned 

We have added a mention of Figure S7 to page 8 in the following sentence (note that we have renumbered our Supplementary Figures to be clearer about which are figures and which are tables, so Figure S7 has become Figure S5 in the new draft): 

“While we have determined the MIC values of the Wylie mutant data set (see the Supporting Information, including Figures S4 and S5, for further information), here, we overwhelmingly employ the more comprehensive set of fitness values acquired by Firnberg et al. in our analyses (see Table 1)\\cite{firnberg2014comprehensive}.”

Reference: 35,41,57,69,70,72,77: Remove DOI number please. 

We have removed these numbers based upon your recommendation. 

Supporting information:

Page 6 : “10^6 cfu/mL, not 106 cfu/mL” 

Thank you, we have fixed the exponent. 

Figure S4. Date of the Wylie mutants are incomplete 

Presumably, you are referring to the fact that we list 12, not 15, single mutants. This is because the table is meant to list only the Wylie double mutants and their constituent singles (constituent meaning only the singles that are part of the double mutants). Only 12 of the 15 Wylie mutants were part of the 6 double mutants (and therefore “constituent single.”. This table is thus complete for the purpose it is meant to serve. 

I found Figure S6 to be confusing. Some Wylie mutants are missing (A172P, G218V, S70G) and there are others mutants (not named in the manuscript) 

The intent of this figure was simply to show that there is a reasonably strong positive correlation between MIC and fitness data for beta-lactamase mutants. Wylie mutants don’t have to be used to determine the strength of this correlation. This is why we left out some Wylie MIC data and included other mutant data (by the way, G218V is in the figure). Ultimately, this plot showed that we can safely use the expansive fitness data set we employed throughout the main text, which should be the focus of one’s attention, not the MIC data set (which is why all MIC data is conscribed to the supplement). To clarify this point, we changed the name of this section of the Supplement to “Correlation Between MIC Data and Firnberg Fitness Data” instead of “Correlation Between Wylie Mutant MIC Data and Firnberg Fitness Data.” . 

We thank the reviewer again for his/her/their careful review of our manuscript, and in particular, for pointing out our various omissions. 

Additional Edits Introduced by Authors During the Review Process

During the review process, we realized that our original docking calculations were performed on ampicillin in the wrong protonation state (with a COOH). Ampicillin has a pKa of 2.5 and is therefore strongly acidic. As such, the correct protonation state to use should be COO-. We thus reran our Autodock Vina calculations with this corrected protonation state. As a consistency check, we also changed our AutoDock Vina grid box size from 40x60x40 to 24x26x24. Because we would like to obtain our docking scores that reflect ampicillin attempting to bind to the active site, we felt that using a smaller grid box would ensure that we do not end up measuring docking scores associated with spurious docking events to non-active site portions of the enzyme. Altogether, these modifications did not change our docking scores significantly. This is apparent from the plots of fitness vs. docking score presented below from before and after our change. 

In the left plot, we plot our new results (our new Figure 10) with the protonated carboxylate (COO-) and the 24x26x24 grid box; on the right, we plot our old results with the carboxylic accid and a 40x60x40 grid box. The overall shape and range of the distribution remains essentially the same, as is also indicated by the comparatively small changes to the fit parameters. We have nevertheless corrected all of the manuscript's plots, including main text Figures 9 and 10 and Supplementary Materials Figure 6, that were based on the original docking calculations. 

We have also added the following text explaining the appropriate ampicillin protonation states to the “Computing Ampicillin Binding Free Energies with Autodock Vina” section: 

“Because of ampicillin's low pKa of 2.5, all ampicillin carboxylic acid groups were modeled as carboxylates in our ADV simulations.”

and

“We chose to employ a relatively small grid box so as to reduce the chance that ampicillin binds to a non-active site region of the enzyme, which is undesirable.”

We furthermore note that we have performed Smina calculations that independently verified the accuracy of our new Autodock Vina results, as evidenced by the plot of fitness vs. Smina binding free energy changes below.

---

## [Decision Letter · Decision Letter 1]

7 May 2020

Predicting the viability of beta-lactamase: How folding and binding free energies correlate with beta-lactamase fitness

PONE-D-20-04028R1

Dear Dr. Rubenstein,

We are pleased to inform you that your manuscript has been judged scientifically suitable for publication and will be formally accepted for publication once it complies with all outstanding technical requirements.

With kind regards,

Jose M. Sanchez-Ruiz

Academic Editor

PLOS ONE

Additional Editor Comments (optional):

Reviewers' comments:

Reviewer's Responses to Questions

**Comments to the Author**

1. If the authors have adequately addressed your comments raised in a previous round of review and you feel that this manuscript is now acceptable for publication, you may indicate that here to bypass the “Comments to the Author” section, enter your conflict of interest statement in the “Confidential to Editor” section, and submit your "Accept" recommendation.

Reviewer #1: All comments have been addressed

Reviewer #2: All comments have been addressed

2. Is the manuscript technically sound, and do the data support the conclusions?

Reviewer #1: Yes

Reviewer #2: Yes

3. Has the statistical analysis been performed appropriately and rigorously? 

Reviewer #1: Yes

Reviewer #2: Yes

4. Have the authors made all data underlying the findings in their manuscript fully available?

Reviewer #1: Yes

Reviewer #2: Yes

5. Is the manuscript presented in an intelligible fashion and written in standard English?

Reviewer #1: Yes

Reviewer #2: Yes

6. Review Comments to the Author

Reviewer #1: (No Response)

Reviewer #2: The authors responded satisfactorily to the prior reviewers' comments, this reviewer is pleased to recommend its publication.

7. PLOS authors have the option to publish the peer review history of their article (what does this mean?). If published, this will include your full peer review and any attached files.

Reviewer #1: No

Reviewer #2: No

---

## [Editor Report · Acceptance letter]

15 May 2020

PONE-D-20-04028R1 

Predicting the viability of beta-lactamase: How folding and binding free energies correlate with beta-lactamase fitness 

Dear Dr. Rubenstein:

I am pleased to inform you that your manuscript has been deemed suitable for publication in PLOS ONE. Congratulations! Your manuscript is now with our production department. 

With kind regards,

on behalf of

Prof. Jose M. Sanchez-Ruiz 

Academic Editor

PLOS ONE